# Histone Deacetylases (HDACs): Evolution, Specificity, Role in Transcriptional Complexes, and Pharmacological Actionability

**DOI:** 10.3390/genes11050556

**Published:** 2020-05-15

**Authors:** Giorgio Milazzo, Daniele Mercatelli, Giulia Di Muzio, Luca Triboli, Piergiuseppe De Rosa, Giovanni Perini, Federico M. Giorgi

**Affiliations:** Department of Pharmacy and Biotechnology, University of Bologna, Via Selmi 3, 41026 Bologna, Italy; giorgio.milazzo@unibo.it (G.M.); danielemercatelli@gmail.com (D.M.); giulia.dimuzio@studio.unibo.it (G.D.M.); luca.triboli@studenti.unitn.it (L.T.); piergiuseppe.derosa2@unibo.it (P.D.R.); giovanni.perini@unibo.it (G.P.)

**Keywords:** histone deacetylases, HDAC, chromatin, epigenetics, epigenomics, HDAC inhibitors, HDACi, gene networks, cancer, phylogenesis

## Abstract

Histone deacetylases (HDACs) are evolutionary conserved enzymes which operate by removing acetyl groups from histones and other protein regulatory factors, with functional consequences on chromatin remodeling and gene expression profiles. We provide here a review on the recent knowledge accrued on the zinc-dependent HDAC protein family across different species, tissues, and human pathologies, specifically focusing on the role of HDAC inhibitors as anti-cancer agents. We will investigate the chemical specificity of different HDACs and discuss their role in the human interactome as members of chromatin-binding and regulatory complexes.

## 1. Introduction

Histone deacetylases (HDACs) constitute a family of proteins highly conserved across all eukaryotes [1]. Their main action consists in removing acetyl groups from DNA-binding histone proteins, which is generally associated to a decrease in chromatin accessibility for transcription factors (TFs) and specific, repressive effects on gene expression [2]. The function of HDACs is therefore that of driving a higher level of complexity in gene regulatory networks by finely tuning transcript levels in all eukaryotic cells [3]. HDACs are critically involved in physiological processes such as development [4] and cellular homeostasis [5], and play an important role in pathological scenarios, such as neurodegenerative disorders [6], genetic diseases [7], and cancer [8]. Understanding the complexity of HDAC function in cells is therefore of paramount importance to design pharmacological strategies to inhibit or modulate their action in the insurgence and sustenance of pathogenesis.

This review aims at providing a comprehensive and updated overview on the current state-of-the-art of HDAC research. We will focus on the transcriptional roles of the largest family of histone deacetylases, zinc-dependent HDACs, while omitting the sirtuins (SIRTs), a family of NAD(+)-dependent histone deacetylases (Table 1) described elsewhere [9]. We will briefly recapitulate the role of histones and histone modifications and provide an evolutionary perspective on the members of HDACs across organisms. We will then describe the specificity of each HDAC family, both in terms of chemical properties, tissue specificity and interacting partners, generating an overview on the current knowledge on the HDAC interactome(s). Finally, we will show the role of HDACs in human pathogenesis, focusing on cancer, concurrently providing an overview on the current drugs adopted in the inhibition and modulation of HDAC activity.

## 2. HDACs in *Homo sapiens* and Other Organisms

HDAC-encoding genes are present in all eukaryotes and likely originated from ancestral acetyl-binding enzymes, which are present across all kingdoms of life [10]: the conservation of their aminoacidic sequence and overall function has allowed scientists to transfer the molecular knowledge between model organisms during the past decades. The first discovery of HDACs dates back the early 1970s, when scientists identified enzymes able to remove acetate from acetate labeled histone solutions in both animal (calf thymus) and plant (spinach leaves) samples [11]. Experiments on human derived cell models, like HeLa cells, quickly followed [12], highlighting sodium butyrate as one of the first HDAC inhibitors. Molecular studies on HDACs continued in many organisms during the 1990s, fully characterizing the two HDACs Hda1 and Rpd3 in the model unicellular eukaryote *Saccharomyces cerevisiae* (budding yeast, [13]) and their role in transcriptional complexes [14]. Molecular knowledge was transferred to mammalian HDACs [15], highlighting a complex network of histone acetylation/deacetylation that involved the fine balance between HATs and HDACs [16].

The interconnection between HDACs and cellular pathways was first discovered in 1997, when it was shown that the overexpression of a histone deacetylase in mouse T-cells led to cell cycle delays [17]. The first human HDAC was identified in 1998 and it was named HDAC1 [18], followed by HDAC2 [19] and HDAC3 [20]. In the following year, three additional human HDAC proteins were discovered: HDAC4, HDAC5, and HDAC6 [21], the latter of which contained two independent catalytic domains. Biochemical and molecular studies on the biology of HDAC have involved several model organisms, including *Caenorhabditis elegans* [22], *Drosophila melanogaster* [23], and *Danio rerio* [24] HDAC-like enzymes have been also shown as regulators of transcription in bacteria, such as the AcuC protein in *Aeromonas hydrophila*, by controlling the acetylation of Acetyl-CoA synthetase which in turn modulates rapid cytoskeletal responses and gene expression [25]. The HDAC family characterization in *Homo sapiens* has been lastly updated to 2002 with the discovery of HDAC11 [26].

Mammalian HDAC proteins are commonly categorized in classes based on sequence similarity to yeast proteins Hda1 and Rpd3 (Table 1). Yeast studies showed that Hda1 plays a more prominent role in regulating the expression of genes involved in carbon metabolite and carbohydrate transport and utilization, while Rpd3 is a master regulator of transcription related to cell cycle progression [27]. Class I mammalian HDACs (HDAC1, HDAC2, HDAC3, and HDAC8) have sequence similarity to Rpd3 [10] protein: a molecule, belonging to Class I HDACs, responsible for the deacetylation of lysine residues on the N-terminal part of the core histones in yeast. The Class II proteins (HDAC4, HDAC5, HDAC6, HDAC7, HDAC9, and HDAC10) have sequence similarity to Hda1 protein, the putative catalytic subunit of the Class II histone deacetylase complex in *Saccharomyces cerevisiae*. Class II is commonly subdivided into two sub-classes (Table 1) based on sequence analysis: IIa (HDAC4, HDAC5, HDAC7, and HDAC9) and IIb (HDAC6 and HDAC10). Finally, the Class IV protein (HDAC11) shares sequence similarity to both Rpd3 and Hda1 proteins. In Appendix A, we show a graphical classification of Zn^2+^-dependent HDACs, setting their color code used throughout the text.

Functional and structural homologs of proper HDAC proteins are the sirtuins, which are also able to de-acetylate histones and in current classifications are also dubbed Class III HDACs (Table 1) [28]. Mammalian SIRTs are seven (SIRT1, SIRT2, SIRT3, SIRT4, SIRT5, SIRT6, and SIRT7) and they possess sequence similarity with the yeast Sir2 protein (Table 1); Sir2 is an HDAC homolog mainly acting a transcriptional silencer [29] for genes involved in amino acid biosynthesis and metabolic pathways [27]. The structural and enzymatic features of SIRT proteins differ significantly from the other HDACs, requiring, for example, NAD^+^ as a cofactor, instead of Zn^2+^ [28]. In the current review, we decided to investigate HDACs belonging to classes I, II, and IV, to be focused in greater detail on the subject of canonical HDACs.

To fully characterize sequence similarity and to propose a model for the evolutionary history of Zn^2+^-HDACs, we performed a phylogenetic analysis based on HDAC protein sequences from selected model organisms (Table 2). The largest amount of sequences (222) derived from eukaryotes, for which we selected a total of 24 species. We added also an HDAC from *Pyrococcus furiosus* [30] as representative of the Archaea kingdom, comprising monocellular organisms that also possess histones and histone modifications involved in transcriptional regulation [31]. Furthermore, to provide an outgroup for our analysis, we included in the analysis three HDAC-like proteins from the Bacteria kingdom, bringing the total analysis to 226 proteins (of which, 223 HDACs, reported in Table 2). The phylogenetic analysis is shown as a maximum likelihood tree in Figure 1.

Our phylogenetic analysis (Figure 1) highlights an early separation of HDACs into the four major classes I, IIa, IIb, and IV. Generally speaking, the number of HDAC genes appears to be proportional with the complexity of organisms, with only higher eukaryotes possessing 11 HDACs. All investigated placental mammals (*Homo sapiens*, *Mus musculus*, *Bos taurus*, and *Sus scrofa*) carry 11 distinct HDAC genes, following the classification reported in Table 1. The marsupial *Monodelphis domestica* (opossum) appears to be missing HDAC7. The complete HDAC family structure is however present in the monotreme *Ornythorincus anatinus* (platypus), suggesting that the HDAC family was already present in the ancestor of mammals. In birds, we noticed that *Gallus gallus* (chicken) is missing HDAC6, observation confirmed in two other selected birds: *Strutio camelus* (ostrich) and *Tyto alba* (barn owl). Reptiles and amphibians possess a mammal-like collection of the classic 11 HDAC paralogs, which are present in *Alligator sinensis* (alligator), *Chelonia mydas* (turtle), *Thamnophis elegans* (snake), and *Xenopus tropicalis* (frog).

In the fish clade, the model organism *Danio rerio* (zebrafish) carries 11 HDAC genes in its genome. A closer inspection however shows that *D.rerio* lacks a HDAC2 ortholog and possesses instead a peculiar HDAC12, which shares homology with HDAC11 (also present in zebrafish as a distinct locus). The same loss of HDAC2 and presence of HDAC12 is observed in *Latimeria chalumnae*, the representative of coealacanths [35], a distinct clade of fish. Strangely enough, we could observe a clear HDAC2 ortholog in the representative of cartilaginous fish *Callorhincus milii* (australian ghostshark), so the loss of HDAC2 and the origin of HDAC12 (probably from HDAC11 duplication) seems to be specific of the coealacanth / ray-finned fish (zebrafish) clade.

Arthropods, while carrying less HDAC genes than vertebrates, possess at least one HDAC per class. *Drosophila melanogaster* (fruit fly) carries a total of 5 HDAC genes, translated in the following protein products: NP_001259507.1 located in Class IIa, NP_001259569.1 in Class IIb, NP_733048.1 in Class IV and two Class I HDACs: NP_647918.2 and NP_651978.2. The malaria-carrier mosquito *Anopheles gambiae* shares a similar HDAC phylogeny as *D.melanogaster*, with five total HDAC genes as well. The scorpion *Centruroides sculpturatus* also possesses the five HDAC1/3/4/6/11 orthologs as the other arthropods investigated, plus an “HDAC7” (XP_023213772.1) highly similar to the scorpion HDAC4 and a bona fide HDAC8 ortholog (XP_023226182.1) (Figure 1).

There appears to be no Class IV HDAC in the genome of the nematode *Caenorhabditis elegans*, which encodes for 8 (3 Class I and 5 Class II) total HDAC-like loci (Figure 1).

Regarding the plant lineage, we selected the model dicotyledon *Arabidopsis thaliana* (thale cress) and the monocotyledon *Oryza sativa* (rice). Both species carry at least one representative for each major HDAC class. *Arabidopsis* in particular carries the largest number of HDAC genes in all species investigated (14, Table 2); this is the result of recent HDAC expansion, as this plant carries a cluster of three recently duplicated Class I HDAC loci, which are located in succession on its genome: NP_190052.1 (encoded by gene At3g44660), NP_190054.2 (At3g44680), and NP_190035.1 (At3g44490). Two more highly homologous *Arabidopsis* Class I HDAC loci, represented by protein NP_198410.1 (gene At5g35600) and NP_201116.1 (gene At5g63110) are separated instead by more than 10 million nucleotides on the plant chromosome 5. Both rice and *Arabidopsis*, as noted before [1] possess unique HDACs that appear to be plant-specific, and that appear in an intermediate position between canonical Class IIa and Class IIb HDACs. Sequences NP_563817.1 (*Arabidopsis*) and XP_015638622.1 (rice), dubbed HDAC8 by the NCBI annotation and in our tree (Figure 1, between Class IV and Class IIb) are even more separated from the rest of the organisms, and appear as a completely unique class of plant-specific HDACs.

Unicellular eukaryotes appear to be lacking a Class IV HDAC member: the *Saccharomyces cerevisiae* (budding yeast) Rpd3 and Hos2 are located within the Class I clade, while Hda1 is located within the Class II clade. *Schizosaccharomyces pombe* (fission yeast) HDACs appear very similar to *S.cerevisiae*, with clear orthologs of Rpd3 (NP_595333.1), Hos2 (NP_594079.1) and Hda1 (NP_595104.1). Finally, the single HDAC reported in the archaeon representative *Pyrococcus furiosus*, represented by RefSeq sequence id WP_011011947.1, appears to be phylogenetically located between Class II and Class IV proteins.

We include as an outgroup a selection of HDAC distant homologs in Bacteria, which we identified in the family of Acetoin utilization proteins (AcuC). We gathered a selection of AcuCs from the Gram-positive *Bacillus subtilis*, the Gram-negative *Escherichia coli*, and *Aquifex aeolicus*, also a Gram-negative bacterium, which provided the first crystal structure of an HDAC-like protein [36]. All these proteins appear to be predating the subdivision in HDAC classes observed in eukaryotes, but are included in our analysis for comparison purposes (Figure 1).

In terms of protein domain architecture, mammalian HDACs highly differ between and within classes, as highlighted by our analysis in Figure 2A, which combines automated [37] and literature-based domain annotations overlaid on the sequence of human HDACs. Each protein is characterized by at least one acetyl-binding catalytic domain, whose localization is conserved within each class. For example, in all the members of Class I and IV, the catalytic domain occupies most of the protein sequence while in Class IIa HDACs this domain is always at the C-terminus of the sequence. Unique amongst its paralogs, HDAC6 has two conserved catalytic domains [38] and a zinc finger domain. Class IIa HDACs are characterized by a catalytic domain occupying the C-terminal half of the protein, and by a N-terminal part rich in interaction domains, including with the MEF2 transcription factor, a master regulator of muscle organogenesis [39,40]. The N-terminal Class IIa HDAC region contains also several serine (S) residues, targets of CaM kinases which when phosphorylated, allow interaction of Class IIa HDACs to the chaperone protein 14-3-3 and export to the cytoplasm [41].

Indeed, HDACs are predominantly located in the nucleus, but some have been observed in the cytoplasm as well (Table 1), such as HDAC6 [42]. For Class I HDACs, subcellular localization is predominantly nuclear and it depends on nuclear localization sequences (NLSs) localized near the N-terminal portion [43,44]. For HDAC1, this NLS is encoded by the sequence KKAKRVKT, located in the C-terminal portion of the protein, and necessary to transfer it to the nucleus [43]. The NLS is also present at the C-terminus of other Class I HDACs: HDAC2 [45] and HDAC3 [46] and in the middle of the catalytic domain of the shorter HDAC8 [47]. 

The Class IIa HDAC NLS is enriched in arginines (R) and lysines (K) and highly conserved across vertebrates (Figure 2B). In Class IIa HDACs, the nuclear localization signal of NLS is balanced by the presence of nuclear exporting signals (NES), a fact that sustains the nuclear/cytoplasm translocations of these proteins [48].

## 3. HDAC Specificity

### 3.1. Enzymatic Specificity of HDAC Activity 

Genomic DNA is compacted in chromatin fibers organized in a highly dynamic pattern of octameric elements called nucleosomes composed by two copies of each four histone proteins H2A, H2B, H3, and H4 [50]. The flexible nature of chromatin, from the ‘beads-on-a-string’ structure to a more compacted 30-nm diameter fiber, is directly associated to the transcriptional rates of all the known genes of higher organisms [51]. The modulation of histone H3 and H4 N-terminal tails by several post translational modifications (PTMs) translates into differential accessibility and activity of transcription regulators on specific DNA sequences. One of the most important histone PTMs in relation with transcriptional regulation is H3 and H4 Lysine (K) acetylation. Lysine acetylation is a reversible transcriptional modification which occurs at highly conserved residues within the N-terminal histone tails, contrasting the intrinsic trend of chromatin fibers to curl up into highly compact structures [52].

Histone acetylation is modulated by the antagonistic, although balanced, enzymatic activity of HDACs and histone acetyltransferases (HATs) [53]. HATs induce a relaxation of chromatin through the addition of an acetyl group to specific lysines on the histone N-tails. The negative charge of acetyl groups leads to a reduced affinity of histones to DNA chromatin thus facilitating access of transcription factors and other transcriptional machinery. On the other hand, HDACs remove negatively charged acetyl groups from histone lysines, leading to a more compacted chromatin and lower transcriptional accessibility [2]. HDACs lack intrinsic DNA binding activity and thus their precise genomic localization is given via the physical interaction with target-specific transcriptional regulators, or through their incorporation into large multiprotein transcriptional complexes [54].

While this review will focus on the role of HDACs in transcriptional regulation, these proteins act on other molecular mechanisms as well, including metabolic processes and DNA metabolism, by virtue of removing acetyl groups from both histone and non-histone proteins [55]. The pathogenic deregulation of several HDAC enzymes is demonstrated to play a key role also in DNA replication, genomic stability, and also DNA damage response [56,57]. For example, valproic acid (VPA) has been shown to counteract double strand break processing in yeast cells by inhibiting Class I HDACs [58]. Even in the field of virology, HDACs have been hinted to play a role. The current worldwide effort to molecularly characterize the interactome of the deadly SARS-CoV-2 (Severe Acute Respiratory Syndrome Coronavirus 2), responsible for the COVID-19 (Coronavirus Disease 2019) pandemic [59] has experimentally validated the human HDAC2 to be the only interactor of the viral Non-Structural Protein 5 (NSP5) in HEK293T cells [60]. NSP5 is a protease with largely unknown roles but essential for viral replication [61]. The physical interaction of NSP5 is predicted to cleave the HDAC2 catalytic domain from the NLS (Figure 2), thereby driving its permanence in the cytosol [60].

As previously mentioned, the first structure of the HDAC catalytic domain was obtained studying the bacterial *Aquifex aeolicus* HDAC homologue, known as Histone Deacetylase-Like Protein (HDLP) [36]. Structural and comparative analyses of the catalytic pocket in HDLP and Class I HDAC8 enabled speculation as to the catalytic mechanism of these enzymes (Figure 3A) [62]. A zinc ion (Zn^2+^) is required as cofactor of the reaction: it is coordinated by three amino acidic residues (DDH) and is involved in the stabilization of the acetyl-Lysine inside the catalytic pocket. The Zn^2+^ cation further polarizes the C-O bound of acetyl group, making the carbonyl carbon a better target (more electrophilic) for the nucleophilic attack by a water molecule which, in turn, is activated by a histidine (H) residue. One tyrosine (Y) residue stabilizes the transition state of the substrate and a histidine residue promotes the definitive removal of the acetyl group.

The amino acids involved in the catalytical process are highly conserved between all canonical HDACs and across species. The degree of conservation of the key HDAC catalytical residues shown in Figure 3A can be appreciated in Figure 3B, where we show sequence conservation logos focused on four regions containing these key amino acids (marked with red asterisks). A remarkable exception is the amino acid tyrosine at position 345, which is replaced by a histidine in Class IIa HDACs.

Class I HDACs show the strongest histone deacetylase activity, while the remaining classes show preference for other substrates [63]. The unbiased identification of specific HDAC substrates can be obtained using two well-documented experimental strategies. The first one studies the variation of cell acetylation status (‘acetylome’) using stable isotope labeling of amino acids in cell culture (SILAC) in the presence or absence of a specific HDAC inhibitor [64]. The second one provides the incubation of mammalian cell lysate with a recombinant HDAC protein carrying a photoactivable amino acid into the catalytic site: irradiation with UV light promotes the crosslinking of recombinant HDAC to putative substrates, which can be identified through HDAC Immunoprecipitation followed by mass spectrometry [65]. These methodologies were applied for the identification of HDAC8 candidate substrates using, respectively, the specific HDAC8 inhibitor PCI-34051 and a recombinant HDAC8 protein in which tyrosine 100 is replaced with a *p*-benzoyl-*L*-phenylalanine (Bpa) [64,65].

#### 3.1.1. Class I: (HDAC1, -2, -3, and -8)

Class I HDACs localize into the nucleus and can deacetylate all four core histones, regulating genome accessibility for transcription [21]. To date, the dissection of the complexity of histone substrate specificity for HDACs has appeared quite challenging for two main reasons. Firstly, each Class I HDAC enzyme can deacetylate several Lysine residues on histone tails, and most of these Lysine residues can be deacetylated by more than one Class I HDAC [66]. Moreover, Class I HDACs mainly operate in protein complex in which more than one HDAC is present. In fact, as described in the next paragraphs, HDACs 1, 2, and 3 generally require association with a multi-protein corepressor complexes for maximal catalytic activity. Several lines of evidence revealed that some HDAC-containing complexes are catalytically activated by inositol tetraphosphate, a small signaling molecule that acts as an ‘inter-molecular glue’ between corepressors and Class I HDACs [67,68,69].

In addition to histones, Class I HDACs deacetylate several nonhistone proteins. Because of their predominant nuclear localization, known substrates of HDAC1 are transcription factors such as: p53 [70], E2F1 [71], the corepressor YY1 [72], Proliferating Cell Nuclear Antigen (PCNA) [73], and lysine demethylase 1 (LSD1) [74]. The effect of the deacetylation of these proteins are, respectively, the destabilization of p53, the inactivation of E2F1, the reduced repressive effect of YY1, the reduced the ability of PCNA to bind to DNA polymerases beta and delta and the enhanced ability of LSD1 to bind histone H3 for its demethylation. Moreover, molecular screens reported MHS6, AIF1, RuvBL1, and CDK1, as HDAC1 substrates, though the biological relevance of such an activity remains unclear [75]. HDAC8 has been shown to deacetylate ERRα, which results in an enhancement of the transcription factor function [76], and SMC3, one of the components of the cohesin complex, the deacetylation of which facilitates renewal of cohesin following its removal from chromatin during prophase or anaphase [77]. HDAC8 substrates like ARID1A, CSRP2BP, MLL2 [64], HSP90AB1, TUBA1A, TRIM28, ACLY, ITGB1, PFKP, and PDLIM1 [65] have been also discovered through unbiased methodology, but the biological significance of the acetylation status of these proteins requires further investigation.

Besides acting as histone deacetylase, HDAC1-3 and HDAC8 catalyze the removal of crotonyl groups from H3 and H4 histone tails, but little is known about the biological significate of these histone PTM [78,79]. A study showed that, similarly to deacetylation, each HDAC decrotonylates several histone aa residues and each residue can be decrotonylated by more than one HDAC. For example, HDAC1 decrotonylates H3K4cr, H3K9cr, H3K23cr, H4K8cr, and H4K12cr in vitro; H3K8cr can be also decrotonylated by HDAC2 and HDAC8.

#### 3.1.2. Class IIa: (HDAC4, -5, -7, and -9)

Class IIa HDACs, as discussed in paragraph 2, can shuttle between the nucleus and cytoplasm (Table 1). The nuclear export of Class IIa HDACs is due to the phosphorylation of specific serine residues which promotes the binding of 14-3-3 proteins [80]. In particular, 14-3-3 proteins, interacting with Class IIa HDACs, could either mask the NLS, or unmask the NES influencing their localization [81].

About the catalytic activity, for vertebrate Class IIa HDACs, the catalytic Tyrosine 345 residue is replaced by a histidine side chain, which is too short to reach into the active site (Figure 3A). Due to the Y-H substitution, the catalytic activity of those enzymes on acetylated lysines of histone tail peptides is very low when compared to that of Class I HDACs. Nonetheless, they can still exert a strong transcriptional repression as well documented for the transcription factor MEF2, and its repressed target genes [82,83,84]. This property is independent from the Y-H catalytic residue as replacement of H with a Y in Class IIa HDACs promotes deacetylation of acetylated histone tail peptides, but it does not impact on transcriptional repression [85]. This finding suggests that the residual catalytic activity of Class IIa HDACs could be not essential for their transcriptional repression effect on canonical HDACs target but rather it depends on the recruiting of other HDACs, like HDAC3. The dispensable role of catalytic activity in the regulation of gene expression is further supported by the existence of the MITR protein, a splice variant of HDAC9 lacking the deacetylase domain which, like the longer isoform, interacts with MEF2 and represses MEF2-mediated transcription [86,87].

However, the identification of the trifluoroacetyl-lysine as a synthetic specific substrate for these enzymes suggests the existence an “H-associated” specificity substrate [85]. According to this hypothesis, HDAC4 was shown to deacetylate the transcription factor RUNX2 to modulating its activity [88,89]. Furthermore, proteins involved in several metabolic pathways in skeletal muscle are targeted by Class IIa HDAC4 catalytic activity: myosin heavy chain (MyHC) isoforms, peroxisome proliferator-activated receptor gamma co-activator 1alpha (PGC1a), and heat shock cognate 71 kDa protein (Hsc70) [90]. Based on that many more undiscovered Class IIa HDAC substrates might exist.

The evidence herein reported reveals a dual nature of Class IIa HDACs, as scaffold proteins and as active enzymes. However, both aspects of these protein need further elucidations and better characterization.

#### 3.1.3. Class IIb: (HDACs -6 and -10)

HDAC Class IIb is limited to HDAC6 and HDAC10. These proteins are mostly cytoplasmic in mammalian cells [91] thus their main physiological target are non-histone proteins. HDAC6 presents two deacetylase domains known as CD1 and CD2 and a C-terminal zinc finger which show a high binding affinity to free ubiquitin or polyubiquitinated protein [92,93]. Preliminary studies suggested that, although both domains were required for deacetylase activity [94], only CD2 appeared to be catalytically active [38]. However, more recent and thorough data have unquestionably shown that both domains are catalytically active with different substrate specificity: while CD2 exerts its activity on any acetyl-Lysine, CD1 shows preference for deacetylation of C-terminal acetyl-Lysine substrates [95]. HDAC6 can stabilize microtubules by deacetylating α-tubulin, and can control p53 activity by targeting acetyl-Lysine 381/382 [42,96]. Another target of HDAC6 deacetylation activity is MutS homologue-2 (MSH2), a protein involved in DNA mismatch repair [97]. Surprisingly, in this peculiar case HDAC6 can act as an E3 ubiquitin-ligase [97]. In vitro experiments also revealed that HDAC6 deacetylates and subsequentially ubiquitinates MSH2 to decrease its stability [97].

Little is known, instead, about HDAC10 substrates. It plays a role in the metabolism of polyamines, molecules involved in several biological processes, and is frequently altered in tumors [98]. HDAC10 exhibits a strong polyamine deacetylase activity but a poor lysine-deacetylase activity when compared to that of its closest paralogue HDAC6 [99]. However, other findings suggest that HDAC10 may have a deacetylation activity on cytosolic proteins such as Hsc70/Hsp70 and be responsible for promoting autophagy-mediated cell survival [100]. Moreover, like HDAC6, HDAC10 can target MSH2 by deacetylating specific lysines to enhance MSH2 activity in vitro [101].

#### 3.1.4. Class IV: (HDAC11)

HDAC11 is the only component of Class IV and is the most recent discovered HDAC [26]. Since substrates of HDAC11 are not well characterized, the biological function of this protein is poorly understood. Recently, it has been found that HDAC11 has a strong lysine defatty-acylation activity other than deacetylation, a characteristic shared with Class I HDACs and several sirtuins [102,103]. HDAC11 deacetylates SHMT2, influencing the ubiquitination and proteasome degradation of the IFN receptor [104].

### 3.2. Tissue Specificity of HDACs

HDACs were shown to modulate gene expression in several different ways, either during embryonic development or in physiological homeostasis maintenance in adult tissues, where the influence of HDACs on proliferation is largely cell-type specific. The existence of many HDAC paralogs in mammal cells raised the question about tissue-specificity and functional redundancy of this class of proteins. Therefore, mouse models carrying specific HDACs gene deletions were produced to investigate function and role of the different HDACs [4]. We used the GTEx data to get a snapshot of HDACs1-11 gene expression across several healthy tissues (Figure 4).

HDAC 1 and 2 are usually co-expressed in adult tissues, often showing redundancy in their functions [106]. This redundancy could have allowed an evolutionary possibility for the loss of HDAC2 in bony fish (Figure 1). As mentioned before, Class I HDACs predominantly show a nuclear location and are ubiquitously expressed in healthy tissues, even if there are a few studies in which cytoplasmic or organelle location of these HDACs were observed [63]. Transcript levels expressed in FPKM (fragments per kilobase of transcript length per million of mapped reads) show that HDAC1 median expression ranges around 50 FPKM in normal samples, being particularly higher in small intestine, spleen, and thyroid tissues. Lower transcript levels are detectable in brain and muscle tissues (Figure 4). At protein level, high expression can be observed in the thyroid gland, gastrointestinal mucosa, bladder, uterine cervix, and bone marrow. Low protein levels are detected in brain and adipose tissue [107]. HDAC2 expression in healthy tissues is generally mild, with testis the tissue type showing the highest expression of this gene. Its expression is very low in liver, pancreas, and whole blood samples. However, HDAC2 expression varies a lot in blood samples and this variation may account to a difference in expression depending on the different blood-cell types included. Protein levels are generally high in most of the tissues, while very low in the liver [107]. 

Since simultaneous deletion of HDAC1 and 2 genes results in early embryonic lethality conditional mutants were produced to investigate their role during development. Both HDAC1 and HDAC2 show a prominent role in regulating neurogenesis, being HDAC2 essential for adult neurons differentiation and survival [108]. HDAC1 and HDAC2 cooperates in regulating cardiac development, adipogenesis, and hematopoietic lineage differentiation [109], while T-cell differentiation relies mainly on HDAC1 [110].

HDAC3 expression is required in many aspects of development and adult tissues physiology. It is most expressed in human healthy adrenal gland, skin, and spleen, whereas it is poorly represented in a few tissues like heart and pancreas (Figure 4). Protein levels are generally higher in brain and bladder, whereas lower in several tissues like lungs, esophagus, liver, ovary, spleen, skeletal muscles, adipose tissue, and bone marrow [107]. During development, HDAC3 has a role in regulating neuronal cell fate and functions, and coordinates lung development. Its loss causes severe defects in bone formation and increased adipogenesis within the bone marrow. In adult individuals, deletion of HDAC3 in hepatocytes causes an increase of adipogenesis leading to hepatosteatosis whereas its loss in the heart cause interstitial fibrosis. A role in the development of T-cells has also been envisaged particularly in the interplay between intestinal epithelial cells and intestinal lymphocytes in orchestrating host defense against pathogens [111].

Among Class I HDACs, HDAC8 is the least expressed in healthy tissues. Its median expression ranges between 2 to about 8 FPKM, being mostly expressed in adrenal and pituitary glands (Figure 1). Very low protein levels are detectable in spleen and soft tissue, while it is not detectable in uterine cervix and spleen. Higher protein levels are present in thyroid, adrenal and salivary glands, pancreas, and bone marrow [107]. Germline deletion of HDAC8 was shown to cause craniofacial defects resulting in prenatal lethality. It is supposed to have a key role in neuronal development, and it is particularly expressed by cells showing smooth muscle differentiation, being essential for smooth muscle cell contractility [109,112].

In contrast to Class I, expression of Class IIa HDACs appears to be more tissue-specific. They have been shown to take part in different developmental and differentiation processes, involving bone and muscle tissues, immune cells, brain, and the vascular system [113]. At transcript level, HDAC4 is generally low in all healthy tissues with few exceptions: a higher expression is in fact detectable in colon, muscle tissues (Figure 4), and bones, showing a central role in skeletal formation and remodeling. Mouse models of HDAC4 deficiency have shown impaired osteogenesis and severe defects in brain development [81,113]. Protein expression is higher in thyroid, lungs, gastrointestinal system, breast, pancreas, cervix, endometrium, and bone marrow [107].

HDAC5 is highly enriched in muscles, heart and brain [4]. As shown in Figure 4, it shows a higher expression in blood vessels, brain tissues, pituitary gland, and skin samples, while it is poorly expressed in liver and pancreas. Knockout mice show exacerbated cardiac hypertrophy after stress resulting in cardiac loss of function, and a decreased axon regeneration ability after injury, suggesting a role in neuronal plasticity [113]. Protein levels are higher in few tissues, like thyroid, colon, skin, and bone marrow. HDAC5 is poorly represented in bladder, heart, and skeletal muscles [107]. 

HDAC7 expression varies among different adult healthy tissues, with most of the samples showing median FPKMs ranging from 30 to about 50. Transcript levels are higher in lung, spleen, and uterus, while brain, liver, muscles, pancreas, pituitary gland, stomach, and testis show lower median expression levels (Figure 4). HDAC7 expression is higher in endothelial cells, controlling vascular development and integrity. It is enriched in thymocytes and pre-B cells, but highly downregulated during maturation in macrophages [113].

HDAC9 shows similar functions as HDAC5, being particularly enriched in brain, heart, and muscles [4]. Its deletions in mouse knockout models has been linked to cardiac defects, mainly depending on deregulation of one of its targets, transcription factor MEF2 [63]. It has been also shown to have a role in the modulation of the response of skeletal muscles to motor innervation [4], and to be essential for the correct function of mature neurons [113]. There is some evidence supporting a role in the regulation of development of immune cells. It is highly expressed in T-regulatory cells, where knockout mice developed an autoimmune phenotype, together with increased proliferation and inflammation [113]. Adult healthy tissues show low transcript levels of HDAC9, with most of the tissues showing a median expression ranging between 0 to about 2 FPKMs. Slightly higher transcript levels are detectable in uterine samples, while esophageal and vessel samples show a broad variability among samples (Figure 4). It is poorly represented at protein level in heart and spleen, while generally higher in the gastrointestinal system, bone marrow, kidney, testis, endometrium, uterine cervix, and breast [107].

Class IIb HDAC6 shows a highly variable pattern of expression among healthy tissues, ranging between from 10 to about 30 FPKMs. Lower levels of expression are detectable in blood samples, muscles, and pancreas, while pituitary gland and testis tissues show higher transcript expressions (Figure 4). Consistently, protein expression is higher in testis, breast, and skin samples, while low to absent in liver, esophagus, spleen, skeletal muscles, and adipose tissue [107]. It is involved in the regulation of cytoskeletal dynamics by deacetylating various substrates, including α-tubulin and HSP90α, and plays a role in protein trafficking and proteasome degradation, cell shape maintenance and migration [114]. However, deletion of HDAC6 led to no evident defects in animal models, possibly due to redundancy of functions shared with HDAC10 [4]. It has been also reported recently that HDAC6 may have a role in neuronal development and function [115].

Very little is known about HDAC10 roles in tissue physiology. It has been described to share functional similarity to HDAC6. Its expression is slightly higher in pituitary gland, prostate, and spleen, while it is poorly represented in blood vessels, heart, and muscles. It is thought to have a role in immune cells development, and it has been recently shown that HDAC10 deletion improves Foxp3+ Treg cells suppressive function in vivo [116]. Protein staining is generally medium to high in all tissues [107].

As stated before, little is also known on the physiological role of Class IV HDAC11, which appears to be entirely missing in monocellular eukaryotes (Figure 1). In human tissues, HDAC11 is particularly enriched in brain and testis (Figure 4), but also heart, kidney, nerves, and muscles [4]. In particular, HDAC11 expression has been reported in immune cells, with roles in repressing anti-inflammatory cytokine pathways [117].

## 4. HDACs in Transcriptional Complexes

HDAC-controlled specific patterns of gene expression are observed not only in different histological types, but also in distinct phases of the cell cycle. The modulation of a specific gene expression pattern—in response to physiological and pathological signals—is due to the action of transcriptional regulators that bind specific DNA sequences and to the modification of chromatin structure, which in turn controls the accessibility of DNA to regulatory factors. Thus, the specificity of HDACs for regulation of distinct expression profiles depends on the cell type and on the different partner proteins typically expressed in that cell, in addition to the signaling context of the cell [4,63].

As explained above, Class I HDACs are nuclear long-lived proteins ubiquitously expressed in human tissues. Both HDAC1 and HDAC2 can interact with each other and with other proteins to generate nuclear protein complexes that can deacetylate specific histone targets. Experimental evidence until now has shown that Class I HDACs can take part to four distinct protein complexes: Sin3, NuRD, CoREST, and MiDac [118]. A schematic representation of the identified HDAC1/2 complexes is shown in Figure 5.

### 4.1. Sin3 Complex

The mammalian Sin3 histone deacetylase is the prototypical co-repressor complex and it is recruited to chromatin by DNA bound repressor proteins to facilitate local histone deacetylation and transcriptional repression [119]. Sin3 protein was initially described in *Saccharomyces cerevisiae* as positive regulator of transcription, and recently many reports have demonstrated that the mammalian Sin3/HDAC complex can also be recruited at actively transcribed genes [120].

Compositionally and functionally, two distinct Sin3 complexes (A and B) are reported in different types of cells. The Sin3 complexes play roles in almost all nuclear processes, as both global chromatin regulators and gene-specific transcriptional co-regulators for the control of chromatin homeostasis throughout the cell cycle. The existence of these two major Sin3 complexes has been already demonstrated in yeast and in mammals. Specifically, it was demonstrated that SIN3A may have a preference for HDAC1 over HDAC2 [121,122]. SIN3A and SIN3B proteins are encoded by paralogous genes and are the most well characterized components of Sin3 complexes in mammalian cells. Both SIN3A and SIN3B proteins have multiple direct protein–protein interfaces (PAH, HID and Sin3a_C domains) and can recruit a variety of cofactors (HDAC1, HDAC2 Tet1, OGT, Arid4A, Ing2, etc.) that compose different Sin3-complexes. The core of the Sin3 complex is the SIN3A/B protein module which binds directly to HDAC1/2 and to other chromatin adaptors and transcription factors [119,123,124]. Both SIN3A and SIN3B complexes combine regulation of individual target genes (Rb-E2F axis, Myc/Mxd network, etc.) with house-keeping roles as global chromatin organizers (DNA replication, peri-centric heterochromatin, etc.) in the progression of (or exit from) cell cycle [119].

The interaction of SIN3A/HDAC1 complex with cell cycle regulators such as Rb and the Mxd1 family suggests that loss of SIN3A would cause an uncontrollably cell cycle progression. This latter hypothesis is supported by a direct correlation between downregulation of SIN3A and a less proliferative potential after treatment of tumoral in vitro cell models treated with HDAC inhibitors, such as SAHA or MS-275 (which specifically target Class I HDACs) [119]. 

The role of SIN3A/HDAC1 has also been demonstrated in the maintenance of male germ cell viability. SIN3A deleted germ cells accumulate DNA damage, that culminates in cell death by apoptosis and the authors linked this phenotype to upregulation of c-Myc genes due to reduced Mxd-family repression via Sin3A and enhanced DNA damage [125]. The complex also plays a critical role during mouse lung development, where it prevents the induction of a senescence-like state of early lung endoderm progenitor cells through transcriptional repression of Cdkn1a (also known as p21) and Cdkn2 (also known as p16), two cell cycle inhibitor factors [126].

Loss of SIN3A complex activity in the early foregut endoderm of the developing mouse resulted in a permanent cell cycle arrest in G1 phase. The G1/S transition block is caused by upregulation of p16 and p21 and, phenotypically, has been represented through profound defect in lung development and progressive atrophy of the proximal lung endoderm with complete epithelial loss at later stages of development [127]. Depending on the dynamics of its spatio-temporal protein interactome, Sin3 complexes, can have a tumor suppressor function, as described earlier, or it can exert oncogenic function with key roles in development and progression of several types of cancer [120]. In acute promyelocytic leukemia it was reported an interaction between the fusion oncoprotein PLZF–RARα, generated by a chromosomal translocation, and the Sin3a complex to repress retinoid-responsive genes, inhibit differentiation and induce cell transformation [128]. In *Drosophila melanogaster*, the interaction between SIN3A and HDAC1 orthologs is conserved [129], and their complex has been shown to be a pivotal and positive regulator of cellular invasion and epithelial-to-mesenchymal transition (EMT) in multiple endocrine neoplasia type 2B (MEN2B) [130].

The Sin3b-HDAC1/2 complex has also been shown to carry out a tumorigenic role in cancer via the establishment of inflammatory statuses. Specifically, in pancreatic cancer, the Sin3b complex promotes cellular senescence [131]. Senescent preneoplastic lesions are formed and secrete SAPS cytokines (senescence-associated secretory phenotype), which in turn promotes immune system activation. Immune cells further promote the generation of a pro inflammatory environment, which induce pancreatic cancer progression [132].

### 4.2. Co Rest Complex

HDAC1 and HDAC2 take part in the formation of the CoREST complex, which also contains Lysine Specific Demethylase 1 (LSD1) and RCOR proteins 1 to 3 (Figure 5). The CoREST complex is a chromatin-modifying co-repressor complex, originally described as a corepressor of REST (RE1- silencing transcription factor), able to regulate neuronal gene expression and neuronal stem cell fate [133]. REST protein can interact with the CoREST complex via its C-terminal domain and with the Sin3a complex via its N-terminal domain. Thus, CoREST and Sin3a can be found together in REST regulated chromatin regions [134]. The RCOR proteins act as scaffold allowing the complex assembly and the recruitment of the entire complex to the repressive transcription factors [135]. Moreover, other subunits, including Sox-like protein, ZNF217, and p80 have been found in association with the complex [136]. 

Recently, structural investigations based on the small angle X-ray scattering (SAXS) methodology, crosslinking-MS, negative-stain EM, and cryo-EM have described CoREST complex as a bi-lobed structure where LSD1 and HDAC1/2 are positioned at either ends of the complex [135]. RCOR1/2/3 mediate the interaction between the two CoREST enzymatic factors: LSD1 and HDAC1/2. In fact, ELM2-SANT1 domain of RCOR1 mediates interaction with the catalytic domain of HDAC1 [137] and, on the other hand, the LINKER-SANT2 domain of RCOR interacts with an extended helical region of LSD1, termed Tower domain [138]. The RCOR-SANT2 domain would also appear to facilitate the association of the complex with chromatin by interacting directly with nucleosomal DNA [139].

The distinctive feature of CoREST consist in the ability to remove both acetyl and methyl modifications through the activity of its demethylase (LSD1) and deacetylase (HDAC1) enzymes. In particular, LSD1 can de-methylate both H3K9me1/2 and H3K4me1/2, yet in the context of CoREST, LSD1 seems to preferentially target H3K4me1/2 [140]. Thus, the CoREST complex binds the histone H3 tail in which K4 is mono- or di-methylated to repress transcription by removal of the activating mark H3K9Ac [133].

In the early 2000s, the BRAF–histone deacetylase complex (or BHC complex) was identified and functionally associated to the repression of neuronal genes [141]. This complex contained HDAC 1 or 2, CoREST1, BHC80, BRAF35, and the recently described histone demethylase protein BHC110 now known as LSD1 [142]. BRAF35 contain N-terminal high-mobility group (HMG) domain which bind DNA in a sequence-non-specific manner [143]. Notably, studies in *Schizosaccharomyces pombe* showed that Lsd1 contains a C-terminal HMG domain (lacking in mammalian LSD1). BRAF35 is now considered one of the main partners of LSD1 and HDACs in the BHC complex [141,144]. Recent studies have shown a central role of RCOR proteins 1, 2, and 3 in the formation of a ternary nuclear complex essential for both the deacetylation and demethylation activity performed by respectively HDAC1/2 and LSD1 [135]. Moreover, many biochemical and molecular analyses have suggested the not fundamental role of BRAF35 in the RCORs-LSD1-HDACs protein complex formation [145,146]. Because of that, it is reasonable to hypothesize that the nuclear core complex including RCORs-LSD1-HDACs proteins is not necessarily associated to the BRAF35 protein. In fact, the BHC complex can be considered as a specific instance, BRAF35-containing, of the CoREST complex.

CoREST was also found associated with Gfi1a/b, homologous zinc finger repressors required for hematopoietic differentiation, via the SNAG repression C-terminal domain of Gfi protein. It has been proposed that Gfi1 a/b can stimulate histone de-acetylation by recruiting the CoREST/HDACs complex at specific gene promoters. After that, LSD1 demethylates H3-K4Me1-2 causing the reversibly transcriptional repression of the gene locus. Finally, the recruitment of histone methyltransferase such as G9a or SUV39H1 and methylation of “repressive” sites like H3-K9 induces a stable long-term silencing of targets through the binding to K9-methyl residues of the heterochromatin protein 1 (HP1) that generate the heterochromatinization of the locus [147].

The CoREST complex is an important epigenetic complex in hematopoietic development. In myeloid cells, the CoREST complex is recruited to chromatin via an interaction of LSD1 with the homologous transcription factors GFI1/1B [148]. Because both an aberrant recruitment of HDACs by oncogenic fusion proteins (RUNX1-RUNX1T1 and PML-RAR) and an altered GFI1/1B expression have been implicated in malignant myeloid cell development. Because of that, targeting the CoREST complex remains of major interest in the development of AML therapeutics [148]. 

ZNF217, another CoREST complex component, is a Krüppel-like zinc-finger protein, that specifically binds the DNA sequence CAGAAY (Y means C or T), a consensus highly conserved in the E-cadherin promoter. Physiologically, ZNF217 recruits CoREST complex on the E-cadherin gene promoter to repress the transcription of downstream targets. An aberrant protein levels of ZNF217 has been reported in many cancer cell lines and may cause unregulated targeting by the CoREST-LSD1 complex, with a profound effect on cancer progression [136].

Together, these mechanisms suggest that the HDAC1/2-CoREST complex coordinates gene expression in early embryonic development. In fact loss of LSD1 demethylating activity in ES cells is matched by a reduction in CoREST levels and it is reflected in an aberrant transcription of 588 genes, including precocious expression of genes coding for brachyury, Hoxb7, and Hoxd8, transcription factors with roles in tissue specification and limb development [146].

### 4.3. NuRD Complex

The nucleosome remodeling and deacetylase (NuRD) complex is a CHD-class complex (Chromodomain, Helicase, DNA binding domain) and is one of the only two known complexes coupling two independent chromatin-regulating activities: ATP-remodeling (Tip60/p400) and deacetylation (HDAC1/2) activities (Figure 5) [149]. 

Recently, the application of a new label-free quantitative mass spectrometry method suggested that NuRD complex is composed by one of the CHD3/CHD4, HDAC1/HDAC2, and MBD3/MBD2 (methylated CpG binding) proteins, three MTA1/2/3 (metastasis associated), six RbAp46/48 (retinoblastoma associated protein), two GATAD2b (p66a) or GATAD2a (p66b) and two DOC-1 (deleted in oral cancer) [149,150].

In murine embryonic stem cells (mESCs), PWWP2A or its paralog PWWP2B were identified as stable associated factors of the NuRD complex. Notably, the physical interaction between PWWP2A/PWWP2B and the deacetylase core of NuRD (MTA1/2/3, HDAC1/2, and RBBP4/7) is mutually exclusive with the presence of MBD2/3 (Figure 5). In addition, p66α/β and CHD3/4 can only associate with the MBD2/3 containing complex. Finally, MBD2/3 and PWWP2A/2B confer to the NuRD complex the specificity of chromatin recruitment. In particular, it has been demonstrated that NuRD complex binds methylated DNA in correspondence to the pericentric heterochromatin containing MBD2 proteins. Instead, MBD3 enables the recruitment of the NuRD complex to enhancer and promoter DNA region of active and bivalent polycomb target genes. Moreover, in mESCs, PWWP2A and PWWP2B are enriched at highly active genes. PWWP2A was correlated to H3K36me3 marked genes and PWWP2B to active promoters and enhancers [151]. The NuRD complex directly interacts with several partners, like LSD1 (also present in the HDAC1/2 CoREST complex), Ikaros, Aiolos, Helios, B cell lymphoma 6 (BCL6), the estrogen receptor α (ESR1), and Oct4/Sox2/Klf4/c-Myc (OSKM) [150].

The localization of the NuRD complex on specific chromatin regions is driven by the presence of specific protein subunits. Specifically, MBD2 and MBD3 mediates NuRD recruitment to methylated or hemi-methylated DNA, respectively. The interaction with BEND325 allows for the association of the NuRD to H3K9me3 enriched pericentric heterochromatin. Furthermore, WDR529 and UpSET30 recruits the complex to promoter regions [152,153]. Once positioned, the activity of NuRD complex HDACs is necessary to repress the chromatin status and repress gene expression [154].

The NuRD complex is a key player in various biological processes, like embryonic development, cellular differentiation, hemato- and lymphopoiesis, tumor growth inhibition, or the general repression of transcription [155]. For instance, in mESCs, it has been reported that after H3K36me3 deposition by SET2 on specific promoters of active genes, NuRD/HDAC complex are recruited to the action of PWWP2A/B and the deacetylation of H3K9ac by HDAC2 facilitates RNA Pol II transcriptional elongation [151]. Furthermore, the NuRD co-repressor complex can induce pluripotency in stem cells through transcriptional reprogramming mediated by the interaction with Mbd3 [156].

Several NuRD complex components have been highlighted as putative tumor drivers. In a recent study on hepatocellular carcinoma (HCC), it has been shown that all NuRD components are upregulated in cancer cells when compared to normal tissue [157]. In fact, the CHD4/NuRD complex promotes cell proliferation, migration, invasion, and colony-forming ability and represses apoptosis in HCC cells. It was also showed that this complex has an impact on the immune microenvironment of hepatocellular carcinoma [157]. This upregulation is inversely correlated with CD8 cell and DC cell infiltration in HCC, the two essential anti-tumor immune responses [157].

Finally, recent studies have revealed that the CHD4/NuRD complex regulates complement gene expression. Specifically, the knockdown of CHD4, an ATPase subunit of NuRD complex (Figure 5), dramatically upregulates C4B expression, a critical component of the complement system, and this can trigger proliferation and tumor progression [157,158].

### 4.4. MiDac Complex

MiDac (mitotic deacetylase) complex was the last HDAC1/2 complex to be identified, first in *C. elegans* and then in *Homo sapiens* [118]. The human MiDac complex (Figure 5) contains HDAC1/2, DNTTIP1 (deoxynucleotidyltransferase terminal-interacting protein 1), the co-repressor protein MIDEAS (mitotic deacetylase-associated SANT domain) which has an ELM2–SANT domain like MTA1, and/or the closely related proteins transcriptional-regulating factor 1 (TRERF1 aka: TReP-132, BCAR2, RAPA) and the zinc finger protein 541 (ZNF541 aka: SHIP1); as they too have similar ELM2–SANT domains to MIDEAS, probably TRERF1 and ZNF541 also interact with HDAC1/2 [118].

The active MiDac complex is a tetramer composed by four copies of each of the component proteins. HDAC1 and MIDEAS form a dimer, analogous to the HDAC1:MTA1 complex, and the N-terminal dimerization domain of DNTTIP1 mediates dimerization of these dimeric complex, forming the full tetramer. DNTTIP1 features a C-terminal region structurally similar to the SKI/SNO/DAC domain. Therefore, in addition to contributing to the assembly of this complex, DNTTIP1 also mediates a chromatin-binding activity that is likely to be important for substrate presentation to the active catalytic site of HDAC1 [67].

Proteomic data have revealed that MiDac is associated with cyclin A, suggesting a role in the cell cycle, but probably, as previously described for other HDAC complexes, is the different co-repressor protein that interacts directly with the HDAC catalytic unit to determine the tissue-specific biological function and, therefore, the different targets of the complex [159]. Specifically, MIDEAS, ZNF541, and TRERF1 are the MIDEAS-like proteins responsible for the highly specific MiDac complex’s action. MIDEAS is upregulated in cells blocked in the mitosis phase by nocodazole drug and is also found associated with cyclin A. ZNF541 is a protein that is associated with spermatogenesis and is found upregulated in testis. In mice, transcripts of TRERF1 were found to be highest in the brain, thymus, and testis [67,160].

### 4.5. SMRT Complex

HDAC3 has been described as a catalytic component of a multi-protein complex named CoR/NR/HDAC3 complex (Figure 5) that also includes nuclear receptor corepressors (CoRs) and a nuclear receptor (NR) [161]. The two principal NR CoRs include silencing mediator for retinoid and thyroid hormone receptors (SMRT or NCoR2) and, its homolog, the nuclear receptor corepressor1 (NCoR1). These proteins recruit HDAC3 through the association of the deacetylase-activating domain (DAD) of SMRT to form the core of the NCoR/SMRT complex [162]. Additional core subunits are the G-protein pathway suppressor 2 (GPS2), transducin β-like protein 1 (TBL1, also known as TBL1X), and TBL-related 1 (TBLR1, also known as TBL1XR1) [163].

The NCOR/SMRT N-termini contain highly conserved autonomous repression domains (RDs), namely, RD1–RD3. GPS2 and TBL1/TBLR1 interact with distinct conserved regions of the RD1 to form a three-way core complex. In addition, two SANT (SW13/ADA2/NCOR/TFIIB)-like domains are located between RD1 and RD2, and HDAC3 directly binds to the deacetylase-activating domain (DAD), composed of a DAD-specific motif and one SANT domain. The histone-interacting domain (HID), containing the other SANT domain, preferentially recognizes hypoacetylated histone tails and synergizes with the DAD to promote histone deacetylation and target gene repression [164]. Crystallography studies have revealed that the interaction between HDAC3 and the SMRT Deacetylase Activating Domain (DAD) is based on the action of an essential inositol tetraphosphate molecule [Ins(1,4,5,6)P4,] requires for both the stabilization and activation of the HDAC-corepressor complex [165]. 

The NCoR/SMRT complex has a fundamental role in preserving cellular identity because it binds the pluripotency DNA loci marked by H3K27 acetylation (H3K27ac) and H4K12 acetylation (H4K12ac) and through the HDAC3 catalytic action induces repression of transcription [166]. Somatic cells can be reprogrammed to pluripotent stem cells (iPSCs) by inducing the expression of exogenous factors OCT4, SOX2, KLF4, and c-MYC (OSKM). It has been demonstrated that NCoR complex antagonize OSKM reprogramming by inducing histone deacetylation at restricted loci including pluripotency loci. For these reasons, many strategies to enhance the reprogramming efficiency are based on using pan-HDAC inhibitors such as valproic acid or trichostatin (TSA) which suppress NCoR/SMRT deacetylation activity [167,168].

The importance of HDAC3’s deacetylase activity of NCoR/SMRT corepressor complex was also reported in the regulation of B-cell development and function. In fact, HDAC3 is involved in the regulation of chromatin structure necessary for Immunoglobulin rearrangement (VDJ) during the B-cell development [169].

It has been demonstrated that the catalytic activity of endogenous HDAC3 requires the interaction with DAD domain of either the NCOR1 or SMRT [170] and, according to their biological function as a complex, NCOR1, SMRT, and HDAC3 have been observed to be upregulated in numerous types of cancer including colon, lung, prostate, and breast cancers. Moreover, HDAC3 RNA expression positively correlates with poor survival and prognosis in most of the tumors mentioned above [171]. The pro-oncogenic function of NCoR-HDAC3 complex was first documented in pro-myelocytic leukemia-retinoic acid receptor-α (PML-RARα), pro-myelocytic leukemia zinc-finger retinoic acid receptor α (PLZF-RARα), and AML1-ETO leukemias. In particular, the PML-RARα leukemia is caused by aberrant expression of PML-RARα fusion protein [172]. Physiologically, the NCoR/HDAC complex is a key regulator of the transcriptional repression mediated by PML-RARα [173]. Although, the production of the aberrant PML-RARα protein causes an increase in the binding affinity between RARα and NCoR/SMRT complex. As a result, corepressors cannot be released by the physiological dose of RA. In this manner, through the reduced sensitivity to retinoic acid-dependent transcriptional activation of target genes involved in cell differentiation, leukemogenesis is facilitated [174].

### 4.6. Class IIa HDACs

It has been recently shown that the N-CoR/SMRT-HDAC3 complex, through the C-terminal region of the RD3 domain (RD3c) of SMRT, can also recruit Class IIa HDACs. As already reported, Class IIa HDACs have low deacetylase activity. Specifically, Class IIa HDACs recruitment in the NCoR/SMRT-HDAC3 complex does not promote their deacetylase activity but seems to have a protein scaffold role. Moreover, Class IIa HDACs allow the recruitment of the NCoR/SMRT-HDAC3 complex in the basic transcriptional machine [175].

Intriguingly, Class IIa HDAC but not Class I enzymes are recruited in RD3c suggesting that one or more specific regions of Class IIa enzymes are responsible for this interaction. Structural studies have confirmed this hypothesis, revealing that the zinc-binding structural subdomain is present only in Class IIa HDACs and can have regulatory and structural roles. Some of the transcription factors involved in the recruitment of Class IIa HDACs through the interaction with its N-terminal region, are members of the MEF2 family, Runx2, calmodulin binding transcription activator, and serum response factor [176]. In the bone morphogenetic protein pathway, the level of Runx2 protein is controlled by the action of HDAC4 and HDAC5. On one hand, BMP-2 signaling stimulates p300- mediated Runx2 acetylation, increasing transactivation activity and inhibiting Smurf1-mediated degradation of Runx2. On the other, HDAC4 and HDAC5 deacetylate Runx2, allowing its Smurf-mediated degradation [175]. 

Furthermore, HDAC4 is involved in neuronal synaptic plasticity and memory formation. The HDAC4 deacetylation activity controls the pivotal gene transcriptional program of the central synapsis, affecting information processing in the brain [177]. In mice, deletion of HDAC4 in the forebrain resulted in the impairment of memory, behavioral learning, and long-term synaptic plasticity [178]. In human, the HDAC4 locus is deleted or mutated in patients with brachydactyly mental retardation (BDMR) syndrome, which is characterized by intellectual disabilities, developmental delays, behavioral abnormalities, and skeletal abnormalities [179]. 

HDAC4 is also involved in ataxia telangiectasia neurodegenerative disease. HDAC4 is physiologically located in the cytoplasm of Purkinje cells, but, in patients with ataxia telangiectasia, HDAC4 is detected in the nucleus of Purkinje cells. The non-physiological nuclear localization of HDAC4 leads to the interaction with the transcription factors MEF2A and cAMP response element binding protein. This pathological interaction results in an altered gene expression program associated with degeneration [180].

### 4.7. Class IV HDACs

Class IV is composed by the sole HDAC11 in *Homo sapiens* (HDAC11 and HDAC12 in fish, Figure 1), which shares similarities with both the HDAC’s in Class I and II. These similarities reside both in sequence homology of the N-terminus catalytic domain and in its Lysine de-fatty-acylase activity, that has been added to the previously known deacetylase activity [104,181]. The predominant biological processes attributable to HDAC11 are relative to controlling the transcriptional rates of genes involved in immune system responses, oligodendrocyte development, and mRNA splicing [182].

HDAC11 is functionally associated with the Survival of Motor Neurons (SMN) complex with essential roles in the spliceosomal snRNP assembly [183]. This multiprotein complex (Figure 5) is found in the cytoplasm of metazoan cells and it is composed of six proteins named Gemins 2-7 and the SMN protein. The SMN complex assembles the spliceosomal small nuclear ribonucleoproteins (snRNPs) that have a key role in physiological development of motor neuron [184]. As has been shown by studies already carried out, HDAC11 appears to have a role in the biogenesis or stabilization of the SMN complex. Several studies on the SMN1 deficiency, on lymphoblasts derived from patients with spinal muscular atrophy, showed intron retention in the U12-type intron from the ATXN10 and Thoc2 genes. It has been demonstrated that HDAC11 downregulation causes a similar splicing defect of the U12-type intron just for the ATXN10 gene. In light of these findings, the authors theorized an indirect role for HDAC11, via the SMN complex, on intron retention or a more specialized role in ATXN10 gene processing [183]. However, the physiological role of HDAC11 in SMN-dependent splicing remains unknown. 

HDAC11 also plays a central role in the DNA replication control mechanism. It is known that, for the correct DNA replication, the origin recognition complex (ORC) must bind at the origin of replication, which in turn, recruits Cdc6, Cdt1 and MCM2-7 proteins. HDAC11 was shown to de-acetylate Lys24 and Lys49 at the N-terminus of the chromatin licensing and DNA replication factor 1 (Cdt1) and affect its proteasomal degradation [185,186]. In fact, the acetylation by lysine acetyltransferases of the same Lysine residues has a protective action against SCF-Skp2 complex and DDB1-Cul4 complex, two ubiquitin E3 ligases. On the contrary, the deacetylation of Cdt1 by HDAC11 leads to ubiquitylation and subsequent degradation [186].

## 5. Pan-Cancer Analysis of HDACs

HDACs expression has been shown to possess low cancer specificity. However, deregulation of HDACs has been reported a role in the development and progression of several cancer types. We used the TCGA and TARGET RNA-seq data to get a snapshot of HDACs1-11 gene expression across several tumors (Figure 6A). Survival analysis was performed as described in material and methods to investigate prognostic values associated with HDACs mRNA levels.

Higher expression levels of HDAC1 in the TCGA dataset are detectable in several tumor subtypes, like CESC, DLBC, LUSC, and THYM (Figure 6A), and correlate with a worse prognosis in KIRC, LGG, LIHC, and READ patients. HDAC1 expression associates with a better overall survival (OS) in THYM patients (Figure 6B). Enhanced expression has been previously reported in several solid tumors, including lung adenocarcinoma, bladder, ovarian, prostate, and gastric cancers, often correlating with a poor prognosis [189]. Protein levels reported in the Protein Atlas showed that moderate to strong nuclear immunoreactivity is detectable in most cancer tissues, while several renal cancers are weakly stained or negative [107]. Silencing or inhibition of HDAC1 was proven to be effective in reducing acquired chemoresistance [190] and aggressiveness [191] in cellular models of ovarian and lung cancers. Conversely, in estrogen-receptor positive breast cancer a high expression of HDAC1 was shown to be a good prognostic factor [192]. In several hematological malignancies HDAC1 expression is a bad prognostic factor, and it has been found to be frequently upregulated in T-cell acute lymphocytic leukemia (ALL), chronic lymphocytic leukemia (CLL), Hodgkin lymphoma (HL), and Multiple Myeloma (MM) [189]. Selective inhibition of HDAC1 was found to be effective in reducing tumor growth in experimental models of B-cell CLL [193], and several selective inhibitors are currently under investigation [194].

In tumor samples, HDAC2 increased expression is detectable in NBL, BLCA, BRCA, lung cancers (both LUAD, and LUSC), and some cervical (UCEC), uterine (UCS), and gastric cancers (Figure 2). Most malignant cells display strong nuclear immunoreactivity in several cancer subtypes [107]. Higher expression is associated to a reduced OS in LUAD, mesothelioma (MESO) and soft tissue sarcoma (SARC) patients, even if caution needs to be taken when interpreting the SARC cohort, since the TCGA dataset collects a variety of soft tissue lesions, with different characteristics. A worse prognosis is associated with lower HDAC2 levels in LUSC (Figure 6B). In solid tumors, HDAC2 overexpression has been reported in liver, ovarian cancer and medulloblastoma patients, frequently associated with a poor prognosis [189]. Shorter survival associated to overexpression has been observed in non-Hodgkin and Hodgkin lymphomas [189]. Selective targeting of HDAC2 has been shown to have anticancer properties in preclinical models of different cancer subtypes, neuroblastoma, medulloblastoma, and gastric and liver cancers in particular [189,193,195,196,197].

Tumor tissues showing higher median levels of HDAC3 gene expression comprise BLCA, CESC, GBM, lung, NBL, TGCT, uterine carcinosarcoma, and DLBC (Figure 6A). Most of cancer cells usually show negative or weak HDAC3 nuclear reactivity in several cancer subtypes. Some urothelial cancers, colorectal cancers, and renal cancers show moderate to strong staining [107]. A poor OS is significantly associated to higher expression in LGG and renal cancer (Figure 6B). High levels of HDAC3 were also reported in Hodgkin lymphoma [189]. Selective inhibition of HDAC may be potentially effective in treating castration-resistant prostate cancer [198], lymphoma [199], and breast cancer [200].

Nuclear staining for HDAC8 protein is moderate in several cancers, while cytoplasmic positivity is present in several malignant melanoma, endometrial, and lung cancers. No staining is generally detectable in colorectal, breast, liver, and testis cancer [107]. HDAC8 overexpression is correlated to a poor OS in renal cancer, NBL, and uveal melanoma, while significantly better outcomes are reported for adrenocortical carcinoma (ACC), LUSC, skin melanoma (SKCM), and stomach adenocarcinoma (STAD) compared to low-expressing patients (Figure 6B). Its expression was found to be increased in BRAF-mutated melanoma. Overexpression of HDAC8 and 3 was associated with an improved survival in stage IV metastatic melanoma [189]. As we highlighted before, HDAC8 is one of the least expressed HDACs in physiological tissues (Figure 4) and for this reason, it has been considered a most promising target to develop anticancer selective therapies where HDAC8 is pathologically upregulated, without severe side effects [201].

Class IIa HDACs are frequently dysregulated in some cancer subtypes. Considering the TCGA data, HDAC4 expression is higher in LGG, THYM, UVM, and acute myeloid leukemia samples (Figure 6A), showing a prognostic significance in ACC, SKCM, and UVM (Figure 6B). Low transcript levels of HDAC4 are correlated with a better OS in LGG, pancreatic adenocarcinoma (PAAD) and NBL. Negative staining is showed in lung and renal cancers, while a mild immunoreactivity is observable in stomach, urothelial, testis, and skin cancers. In several cases of most of cancers moderate to strong cytoplasmic immunoreactivity is displayed, with few cases also showing additional nuclear staining [107]. Compared to normal counterpart, HDAC4 has been also reported to be significantly upregulated in gastric cancers and T-cells ALL [189]. Low transcript levels compared to normal counterpart can be detected in bladder, colorectal, and uterine tumors [189].

HDAC5 shows most prominent transcript levels in LGG, NBL, PAPG, and UVM (Figure 6A), while lower expression is detectable in gastric cancers (ESCA, STAD), lung cancer, and OV. Its downregulation correlates with a worse OS in renal cancers (Figure 6B). Upregulation of HDAC5 was also reported in hepatocellular carcinoma tissues [189] and breast cancer [202], while a lower expression is reported in urothelial cancer and acute myeloid leukemia [189]. Protein staining is usually negative in head and neck, several cervical, ovarian, and gastric cancers. Weak to moderate cytoplasmic positivity is detectable in glioma, thyroid, prostate, lung, and liver cancers, while positivity is frequently observed in colorectal, testis and breast cancer [107].

Compared to normal counterparts, HDAC7 shows downregulation in several different subtypes, like breast carcinoma, head, and neck cancers; renal, colorectal and lung tumors; prostate cancers; and uterine carcinomas. In tumor samples, increased expression of HDAC7 can be observed in pancreas and testicular tumors (Figure 6A). Survival analysis reported several significant associations in different tumor subtypes: high transcript levels are associated with a reduced OS in CESC, GBM, LGG, and KIRP, while it is a good prognostic factor in BLCA, DLBC, KICH, NBL, and THYM (Figure 6B). Overexpression has been previously reported to be associated with poor prognosis in pancreatic tumors. An association with tumor aggressiveness and prognosis has also reported in gastric [203] and ovarian cancers [204]. Overexpression of HDAC7 is frequently reported in several hematologic malignancies like ALL and CLL, often correlated with poor outcomes [189]. 

HDAC9 expression is generally low in tumor samples, with some subtypes showing higher transcript levels like PCPG and LALM (Figure 6A), while downregulation can be observed in colorectal adenocarcinomas. Upregulation of HDAC9 can be detected in esophageal carcinomas, renal cancers (KIRC and KIRP), non-small cell lung cancer, and thyroid cancer. Low expression is associated with a significantly poorer OS in ACC and CHOL. Upregulation of HDAC9 was reported in medulloblastoma, where its depletion was found to be effective in increasing cell death [189]. Upregulation was detected in B-lineage ALL and CLL, associated with reduced life expectancy and more advanced disease stages [189]. At protein level, most cancer cells display moderate to strong nuclear immunoreactivity with additional cytoplasmic staining in many cases, being particularly higher in colorectal, head and neck, breast, cervical, ovarian, and skin cancer. Renal, stomach, liver, prostate, and endometrial cancers show generally weak or negative immunoreactivity [107].

Class IIb HDAC6 is downregulated in several tumors, like prostate and stomach adenocarcinomas, renal tumors (KICH), and uterine carcinoma (UCEC). Compared to normal tissues, HDAC6 shows higher median FPKMs in different tumor subtypes, like neuroblastoma, low-grade glioma, and pancreatic adenocarcinoma (Figure 4 and Figure 6A). Increased expression was also reported in bladder cancer, malignant melanoma, and lung cancer [205]. In the majority of cancer tissues, weak to moderate HDAC6 immunoreactivity is detectable [107]. Downregulation at gene level has been correlated with a reduced life expectancy in adrenocortical, bladder, and pancreatic carcinomas, mesothelioma, and uveal melanoma (Figure 6B). In estrogen-receptor positive invasive ductal carcinomas a better prognosis has been observed in patients having higher HDAC6 expression levels, while low levels are a bad prognostic factor in liver transplantation patients, where HDAC6 downregulation has been linked to an aggressive and neoangiogenesis-promoting phenotype [189]. A frequent deregulation at gene level has been observed in hematologic malignancies, being HDAC6 upregulated in ALL, AML, DLBCL, with a worse impact on prognosis. Its role in CLL is controversial, while some authors report of better outcomes in patients showing higher levels, some others observed a positive correlation of HDAC6 expression with advanced disease stage [189]. Selective inhibition of HDAC6 gave promising results in several cancer models [205,206].

Moderate to strong immunoreactivity to HDAC10 is showed by several malignant cells in different cancer subtypes [107]. It has been found in cervical cancers as metastasis suppressor [181], and low expression is associated to a bad prognosis in lung and gastric cancers [189]. In the TCGA cohort, low expression is associated with a bad OS in BLCA and PCPG, while better outcomes are expected in HDAC10 low-expressing KIRC and THCA patients (Figure 6B). High expression of HDAC10 has been also reported in stage 4 neuroblastoma patients, correlating with poor OS. Being involved in the regulation of autophagic processes and resistance to cytotoxic drugs, HDAC10 inhibition in neuroblastoma cells have been shown to be effective in increasing drug sensitivity [189].

Very little is known about Class IV HDAC11 role in cancer. Its median transcript levels are generally low in all tumor samples, being higher in certain cancer subtypes like KICH, KIRP, LGG, PCPG, and UVM (Figure 6A). Low-expressing LUAD, NBL, and UVM patients experience worse OS (Figure 6B). A key role in the regulation of progression and survival has been reported in NBL [207].

## 6. HDAC Inhibitors as Anticancer Drugs

### 6.1. HDAC Inhibitors

Given the widely reported importance of HDACs in both normal and tumor tissues, a plethora of chemical agents were already identified as HDAC inhibitors (HDACi) and many others are currently under investigation [63]. Based on the huge heterogeneity of the HDAC world, in terms of both putative protein complexes and target substrates, there is a full-blown uneven cellular response to both the specific or nonspecific HDAC inhibition [181,208]. While this paragraph and its classification system will focus on the prominent role of HDACi as anticancer drugs, there is increasing evidence that these compounds can be used to inhibit HDACs in other pathological conditions, such as neurodegenerative, metabolic, cardiac, and immune disorders [209], as well as for targeting HDACs of parasitic micro-organisms [210].

HDAC inhibitors are synthetic or natural cytostatic molecules that can be classified based on their chemical structures and class/subclass specificity. All the chemical compounds designed to inhibit HDAC activity can be broadly subdivided in two main classes: (A) HDAC-specific inhibitors; and (B) HDAC-pan inhibitors. Furthermore, regardless of the specificity, HDACi are also classified into four main groups depending on their chemical structure: (I) hydroxamic acids (hydroxamates), (II) short chain fatty (aliphatic) acids, (III) benzamides, and (IV) cyclic tetrapeptides [189,194,211].

The family/class/HDAC specificity and the types of cancer/disease targeted of all the HDACi herein discussed are summarized in Table 3.

### 6.2. Group I—Hydroxamic Acids

The hydroxamic acid group is composed by several specific and aspecific HDAC inhibitor compounds: Trichostatin A (TSA), Vorinostat (SAHA), Belinostat (PXD101), Panobinostat (LBH589), Givinostat, Resminostat, Abexinostat, Quisinostat, Pracinostat, Rocilinostat, and CHR-3996. The rationale behind the effectiveness of hydroxamic acids in the inhibition of canonical HDAC enzymes (Table 1) lies within the ability to chelate the Zn^2+^ cation in the active site cavity of histone deacetylase enzymes belonging to Class I, IIa, IIb, and IV [63,208].

#### 6.2.1. PAN-Hydroxamic Acid Compounds

In the 90’, the natural product Trichostatin A (TSA) from *Streptomyces hygroscopicus* and the synthetic hybrid polar compound SAHA (Vorinostat) were first identified as reversible hydroxamic acid pan-HDAC inhibitors [212,213,214].

The importance of looking at the Pan-HDAC inhibition to find a way to treat cancer patients is confirmed by the increasing number of positive results obtained in both in vitro and in vivo studies conducted on several types of cancer. These findings are supported by a great number of scientific data of the effectiveness of pan-HDAC inhibition in tumor growth inhibition on a huge number of tumor models including: pancreatic, esophageal squamous cell carcinoma (ESCC), multiple myeloma, prostate carcinoma, gastric cancer, leukemia, breast, liver cancer, ovarian cancer, non-Hodgkin lymphoma, and neuroblastoma (Table 3).

In fact, it should be noted that some of the HDAC pan-inhibitors of the group I were already FDA approved (TSA, SAHA, PXD101, LBH589, and Pracinostat) and almost all the remaining drugs are currently in clinical trial phases I–III for many types of cancer [208]. The majority of pan-HDAC inhibitors have been reported to interfere in the cell cycle progression by blocking G1/S_G2/M transition. Cell cycle blocks is mainly caused by the mis-regulation of key genes such as CDKN1A and AKT and by the hyperacetylation/activation of the tumor suppressor p53 [215,216]. The effectiveness of pan-HDAC inhibition in blocking tumor cell growth and proliferation was also largely linked to a strong upregulation of key apoptosis related genes such as BAK1, BAD, and BIK [217].

Ideally, the chemical pan-inhibition of HDACs should trigger a global hyperacetylation and, consequentially, an hyperactivation of many genes across the genome. However, many studies have been shown how HDACs pan-inhibition goes along with an important transcriptional repression rather than hyperactivation. Recently, histone H3/H4 acetylation status was investigated on hepatocarcinoma HepG2 cells treated with the pan-HDAC TSA inhibitor. Surprisingly, ChIP-chip analysis has revealed that hyperacetylation of TSS regions seemed to be transient while a strong reduction in the acetylation status was marked over the time and specifically localized in most of the TSS relative to genes downregulated by TSA inhibitor [218,219,220].

Vorinostat (SAHA) is one of the pan-HDAC inhibitor FDA-approved in the management of CTCL (Table 3) and is still under investigation for many others types of cancer [208]. In vitro and in vivo studies have revealed an important link between ARID1A mutation status and SAHA sensitivity in ovarian cancer. Molecular and biochemical data indicated ARID1A expression as necessary condition for the EHZ2-HDAC2 physical interaction that, in turn, lead to cell growth and apoptosis inhibition via phosphatidylinositol 3-kinase (PI3K)/ AKT signaling. Finally, SAHA treatment of several ARID1A KO compared to WT ovarian cancer cells have revealed an important decrease in terms of IC_50_ value of approximately 100 fold [221].

More recently, Jia and colleagues have revealed an important in vivo response of Rb1/Trp53/Crebbp-deficient SCLC cells to the pan-HDAC inhibitor Pracinostat. The physiological role of Crebbp/Ep300 is to promote transcription activation of tumor suppressor adhesion-related transcripts, including CDH1, via acetylation of H3K27 residue of cis regulating elements. The loss of Crebbp leads to reduced H3K27Ac and transcriptional downregulation of CDH1 which, in turn, promotes cell transformation. Pracinostat treatment of DMS53 (human SCLC cells with CRISPR-generated CREBBP deletion) resulted in a widely increase in H3K27Ac, H3K18Ac, and increased CDH1 RNA and protein expression. Moreover, Pracinostat treatment of Rb1/Trp53/Crebbp mice have shown an exceptional therapeutic efficacy investigated by MRI scan of lung tumor burden [222].

Panobinostat (LBH589) is one of the FDA pan-HDAC inhibitor currently used in many phase I/II clinical trials targeting several types of cancer [194]. In vivo mouse xenograft studies of LBH589 treatment on MLL-rearranged acute lymphoblastic leukemia (ALL) cells have shown an important anti-leukemia effect. Specifically, LBH589 treatment seems to induce a depletion of histone H2B ubiquitination via misregulation of the RNF20/RNF40/WAC E3 ligase complex axis [223].

In the last 10 years, because of the massive effect of pan-HDAC inhibition, witnessed by the very low dosage concentration used and to the countless biological functions affected, many scientists have pointed out their attention on the combining the less specific HDACi treatment with other more specific anti-cancer drugs. In neuroblastoma, where the amplification of MYCN interplays with HDACs [224], the p53 network [225] and large scale enhancer complexes [226], the in vitro and in vivo efficacy of the combination treatment with the pan-HDAC inhibitor LBH589 and the BET bromodomain JQ1 compound has recently been demonstrated [227].

#### 6.2.2. Specific Hydroxamic Acid Compounds

Despite most of the hydroxamic acid HDACi being considered as pan- inhibitors, CHR-399 and Rocilinostat were proven to inhibit respectively HDAC Class I and II [194]. In 2010, Moffat and colleagues [228], thanks to docking studies on Class I protein HDAC8, designed and analyzed 23 N-hydroxypyrimidine-5-carboxamide compounds as potential HDAC inhibitors. Pharmacokinetic, viability/proliferation assays and in vivo studies have indicated the 21r compound, also named CHR-3996, as a very good candidate in terms of cell growth inhibition at orally driven nanomolar concentration [228]. Later, Wee and colleagues published the first Rocilinostat, combined with lenalidomide and dexamethasone, phase 1b clinical study on relapsed or refractory multiple myeloma. Results from this combined therapy of the specific HDAC6 inhibitor Rocilinostat, administered orally, have shown positive indications of efficacy and tolerability in 38 multiple myeloma patients [229].

### 6.3. Group II—Short-Chain Fatty (Aliphatic) Acids

Short-chain fatty acids (SCFAs) are broad-spectrum HDAC inhibitors that negatively affect deacetylation ability of Class I and IIa HDACs. Sodium butyrate, phenylacetate, valproic acid (VPA), and phenylbutyrate are the most studied SCFAs [230]. Sodium butyrate is a short (4-carbon atoms) fatty acid compound mainly produced by the intestinal microflora fermentation and represent one of the most important energy source for the mucosa [231]. Moreover, in addition to its physiological role, sodium butyrate was the first SCFA HDAC inhibitor identified in late 70’. In fact, scientists have demonstrated the ability of this SCFA in the inhibition of hyperacetylation of non-H1 histones purified from hepatoma tissue culture models [232]. Specifically, authors have observed that sodium butyrate-induced acetylation of pre-existing non H1-histones was quite rapid, suggesting a direct role of SCFA compounds in the biology of histone acetylation/deacetylation [233,234]. Sodium butyrate and phenyl butyrate compounds, often in combination with other drugs, were tested in a number of phase I–III clinical trials for different types of tumor such as: leukemia, lung cancer, lymphoma, multiple myeloma, and prostate cancer [232].

In vivo studies on the combined treatment with DNMT1 (5-aza cytidine) and HDAC (butyrate) inhibitors on breast cancer have highlighted the intrinsic susceptibility of cancer stem cells (CSCs) to epigenetic targeting drugs. Specifically, authors have shown that 5-aza cytidine/butyrate treatment leads to strong reduction of CSC population in both tumorspheres and in vivo mouse models. Moreover, RNA-seq analyses performed on 3D cell cultures generated from metastatic CSCs (Lin- CD49+CD24+) treated with 5-aza cytidine/butyrate have revealed a mis regulation of cell cycle (Arap1), cell division (Ndc80), and DNA double strand break repair/homologous recombination (RAD51AP1) related genes [235].

Among all the SCFA inhibitors, valproic acid is the most relevant naturally derived short chain fatty acid HDAC inhibitor compound. VPA is a specific HDAC Class I inhibitor and despite being less effective than hydroxamic acids, in terms of de-acetylation inhibition, it is already employed in clinical trial phase I-III for several types of tumor such as: colorectal, prostate, breast, melanoma, NSCLC, and pancreatobiliary [236]. Valproic acid is reported to act specifically against HDAC2 by both inhibiting its deacetylating activity and inducing the expression of E2 ubiquitin conjugating enzyme Ubc which, in turn, leads to HDAC2 proteasomal protein degradation [237,238]. Moreover, transcriptomic analyses on pancreatic cell models have confirmed the ability of VPA treatment in modify the epigenetic status (H3K9Ac) of *cis* regulatory elements of genes involved in cell cycle (cyclin D2) and metastatic cell behavior (Ccnd2) [239,240].

### 6.4. Group III—Benzamides

Benzamide derivative compounds are considered, unlike the majority of the hydroxamate drugs, specific HDAC Class I and IV inhibitors and are reported to be less effective in terms of deacetylation inhibition. Mocetinostat (MGCD0103), Entinostat (MS275), Chidamide (HBI-8000), K560, and K560(1a) are the best characterized HDACi benzamide compounds able to interfere in tumor cell growth of many types of tumor [208]. Mocetinostat and Entinostat are currently under pre-clinical investigation for several types of cancer and have already been taken to clinical trial phase I - II for leukemia, lymphoma, pancreatic cancer, Hodgkin’s lymphoma, SHH medulloblastoma, and other solid tumors [241]. A study has recently highlighted the importance of Mocetinostat treatment in repressing EMT, normally induced by the transcriptional repressor ZEB1 [242]. Specifically, biochemical and molecular analyses on pancreatic cancer cell models have shown an upregulation of ZEB1 target genes and a downregulation of ZEB1 protein level. Moreover, Mocetinostat treatment leads to an increase of the positive H3K4Me3 histone mark on upregulated ZEB1 target genes, suggesting the role of Mocetinostat in the repression of the JARID1 family of histone H3 Lysine 4 demethylases [242].

Recently, in vitro and in vivo studies have suggested that the combined treatment of Mocetinostat HDAC inhibitor and PD-L1 antibody agonist can synergistically exert an antitumoral activity by the modulation of the immune-related gene regulation in lung and renal cancer models [243,244]. In vivo studies of Entinostat treatment on both immunocompetent and immunocompromised ovarian cancer mouse models, respectively C57BL/6 and Rag1 knockout, have revealed a reduction of tumor growth and a prolonged survival only in C57BL/6 mouse model. Interestingly, Entinostat stimulate MHCII pathway only in the immunocompetent C57BL/6 mouse model, suggesting a strong coordination between the epigenetic modulation exerted by MS275 treatment and the consequent stimulation of adaptive immunity. Moreover, flow cytometry and molecular analyses have shown an increase of active intra-tumoral CD8-positive cells together with an substantial activation of IFN-inducible genes such as CXCL10, MHCI, CIITA, and PD-L1 only in the C57BL/6 mouse model [245]. The antitumoral activity of Entinostat treatment has been tested on colorectal cell line models: SW48, HT-29, and Colo-205. Specifically, cytofluorimetric and biochemical assays have revealed that MS275 and 5-fluorouracil co-treatment exert a synergistic effect triggering apoptosis via deregulation of key cell cycle related genes such as p53, CDKN1A, and cyclin A [246].

A recent study on the benzamide Entinostat inhibitor, tested on a triple negative breast cancer (TNBC) cell model, have revealed a good grade of antitumoral activity. Authors have showed that entinostat and SGI (DNMT1 inhibitor) exert a synergist effect in the epigenetic reprogramming of the EMT. Specifically, biochemical and molecular analyses on entinostat and SGI co-treated TNBC cells have revealed a consistent inhibition of WNT signaling and EZH2 expression and an important induction of E-cadherin expression, apoptosis, and the H3K27Me3 repressive histone mark [247].

In two recent studies, two putative HDAC1,2 benzamide specific inhibitors, K560 and K560(1a), were developed and tested for their neuroprotective abilities [248,249]. Authors have shown how K560 benzamide drugs can exert neuroprotective abilities in MPP+ induced toxicity on in vitro SH-SY5Y retinoic acid differentiated cells. Specifically, K560 treatment stimulates HDAC1,2 protein expression and abrogates the cell death effect of MPP+ by modulating key apoptosis-related factors such as claspin, XIAP, and livin, and observed an increased p53 activation through phosphorylation [248,249].

### 6.5. Group IV—Cyclic Peptides

Cyclic peptides are HDAC inhibitor compounds characterized by the most complex “cap” chemical element able to confer a high level of HDAC target specificity [250]. Based on their high spectrum, in terms of actionability on different HDACs, cyclic peptides represent an excellent opportunity for the selective modulation of deacetylation activity inside the cell [251]. Cyclic peptides can be in turn subdivided into two main classes based on the chemical identity of their ZBG group: (1) disulfide-containing bicyclic depsipeptide, or (2) α4-cyclotetrapeptides [251]. 

Bicyclic depsipeptides are considered prodrug compounds which must go through intracellular chemical reactions to expose the ZBG hindered by the homoallylic thiol-containing element [252]. Crystal structure of purified HDAC8 complexed with the bicyclic depsipeptide Largazole have confirmed the ability of the reduced homoallylic thiol-containing element in chelate Zn^2+^ inside the enzymatic active site [253,254].

All the sulfur-containing depsipeptide are secondary metabolites of bacteria and cyanobacteria strains. Specifically, the best characterized HDACi Romidepsin (FK-228), Chromopeptide A, FR901375, Largazole, and Spiruchostatin A are respectively produced by *Chromobacterium violaceum*, *Chromobacterium* sp. HS-13-94, and *Pseudomonas* spp [251,255]. Amongst these, Romidepsin is a prodrug, isolated from *Chromobacterium violaceum*, chemically transformed into an HDACi active molecule by the intracellular glutathione reductase. Consequently, many scientists have demonstrated both in vitro and in vivo its strong cytotoxic activity against several types of cancer [252,254]. Along with that, in 2010 Romidepsin becomes the unique cyclic peptide HDACi inhibitor FDA approved for cutaneous T cell lymphoma (CTCL) [256]. Recently, in vitro and in vivo studies of Romidepsin treatment on HCC have revealed a significant tumor suppression growth via induction of G2/M phase arrest and apoptosis signaling. Specifically, cdc2 phosphorylation and akt Serine 473 dephosphorylation after Romidepsin treatment leads to an increase of p19, p21 and p27 tumor suppressor expression that determine a block of cells in the G2/M transition. Moreover, Romidepsin treatment leads to JNK phosphorylation signaling to induce apoptotic events [257,258].

Consistently, in vitro and in vivo studies on germ cell cancer (GCC) revealed G2/M transition blocks and apoptosis induction after romidepsin treatment. In fact, transcriptomic and H3 acetylated ChIP-sequencing analyses have revealed how the romidepsin treatment leads to a strong deregulation of well-established cell cycle and apoptosis related genes such as: GADD45B, ATF3, FOS, ZFP36, DUSP1, ID2, and CDKN1A [259].

Many more natural bicyclic depsipeptides and macrocyclic peptides exist. Among these, FR901375 chromopeptide A, FR901375, largazole, spiruchostatin A, HC-toxin, trapoxin, and azumamide are currently investigated for their anti-tumorigenic potential [252]. Trapoxin in particular appears to have a high affinity with HDAC8 [260] and has been shown to be inducing stemness genes (such as NANOG, SOX2, and TERT) in adipose tissue derived mesenchymal cells in vitro [261].

A second class of macrocyclic peptides is chemically defined by a cyclic scaffold of D- and L-amino acids [262]. The cyclic tetrapeptide apicidin is a reversible HDACi derived from a fungal metabolite originally isolated from two Fusarium species. In vitro studies on several tumor cell models have revealed antitumorigenic activity of apicidin via induction of cell cycle inhibitor (CDKN1A) and apoptosis (FAS) related genes [263]. Recently, an indirect correlation between apicidin treatment and HDAC8 expression has been highlighted on the oral squamous cell carcinoma model AT-84 [264]. 

## 7. Conclusions and Perspectives

In the pages of this review, we have assessed and shown the enormous complexity of HDACs across evolution, tissues, functions, and interactions. We are however convinced that, even 20 years after the discovery of the last characterized human HDAC [26], the current knowledge still covers only a fraction of the roles of HDACs in the fine tuning of cellular processes. HDACs operate in a still poorly investigated layer of gene regulation that involves chromatin remodeling, enhancer regions and functional elements of the eukaryotic genomes, that all together form a vast, evolutionary conserved, mega-network of protein–protein, protein–metabolite, and protein–DNA interactions. To be able to pharmacologically modulate parts of this network, for example with HDAC inhibitors, investigators will require holistic, systems level investigational mindsets. This will allow to predict and minimize the inevitable undesired side effects that influencing a regulatory network of this proportion will generate, for example in terms of pharmacological toxicity. Fortunately, the high conservation of HDACs across evolution allows for the use of animal models, even outside the mammalian clade (e.g., in Drosophila and yeast), for understanding further functional mechanisms and testing potential inhibitors of these enzymes.

HDACs play a pivotal role in a large system of protein complexes that promote and regulate gene expression, and fundamental steps should be taken in increasing our understanding of their function in all physiological and pathological contexts. The interaction network in which HDACs participate (Figure 5) still lacks a complete structural characterization, which could be achieved dynamically by the implementation of cryomicroscopy on live HDAC complexes [328]. Understanding the functional structure of HDAC-containing transcriptional complexes could allow to assess their heterogeneity and plasticity in components, tissue activity and developmental phase presence. More importantly and ambitiously, HDAC complex characterization will allow the design of novel drugs targeting not their individual components, but the interaction surfaces between the protein subunits. The inhibition of entire HDAC-containing complexes is currently a possibility, as shown for the CoREST complex and specifically via the Corin drug, which inhibits both the LSD1 subunit and HDACs [329].

Further applications of HDAC inhibitors can be found through the activity of these enzymes on non-histone targets, and beyond their canonical functions in transcriptional regulation. For example, inhibition of HDACs could modulate their action in genomic instability, often observed in cancer in the form of amplifications/deletions, chromosomic rearrangements, and chromothripsis [330]. HDACs were also experimentally linked to the pathogenesis of the novel coronavirus, SARS-CoV-2, which has been proposed to trigger the activation of inflammation and interferon pathways through the induction of the cytosolic localization of HDAC2 [60,331].

Further research will provide clinicians with a higher plethora of specific inhibitors, targeting all the chromatin components responsible for tumorigenesis, tumor progression, and tumor maintenance. The definition of specific overexpressed HDACs and transcriptional complexes in a specific tumor, through fast transcriptome or proteome screenings, could allow the adoption of personalized medicine approaches of HDAC activity inhibition.

## Figures and Tables

**Figure 1 genes-11-00556-f001:**
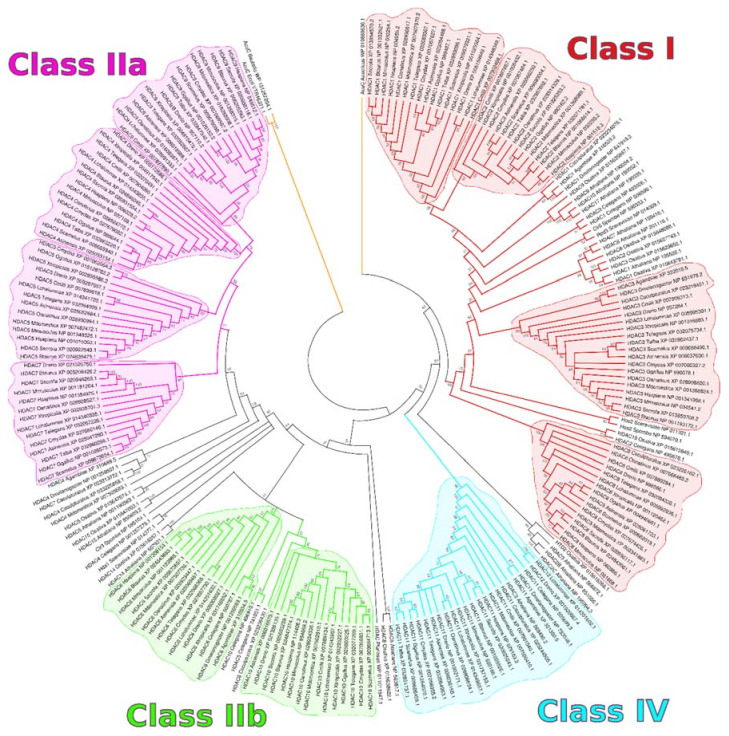
Topological phylogenetic tree representation of 226 representatives of the HDAC protein family. The longest RefSeq protein isoform was selected for each separate gene locus. Multiple sequence alignment was performed using the MUSCLE algorithm [32]. Evolutionary distances were computed as the number of amino acid substitutions per site using the Poisson correction method [33]. The implementation of these algorithms and the visualization were achieved through MEGA X [34]. All 226 sequences used for the generation of this figure are available as Appendix A, in FASTA format. Coloring of branches indicate the putative HDAC class: red for Class I, magenta for Class IIa, green for Class IIb, and cyan for Class IV. Colored areas delimit clades associated to each one of the 11 human HDACs.

**Figure 2 genes-11-00556-f002:**
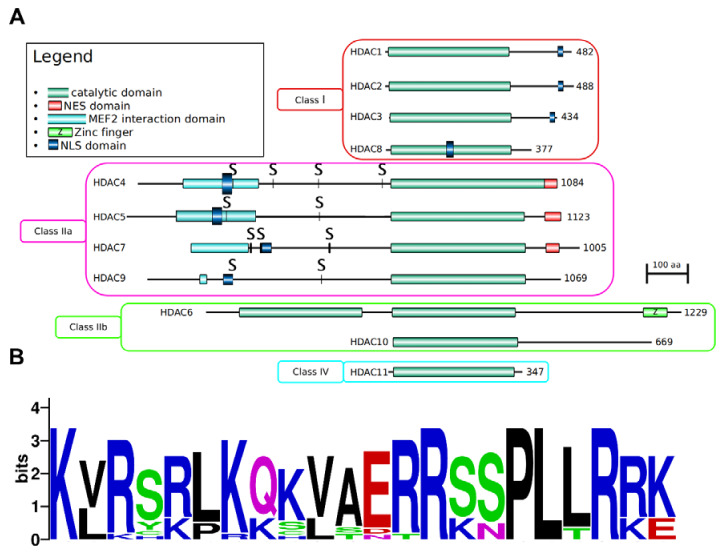
(**A**) Domain architecture of HDAC proteins (human HDACs are used as reference). S indicate Serine residuals phosphorylated by CaM kinases [41]. (**B**) Sequence logo of Class IIa HDAC NLS sequence across human, mouse, chicken, and zebrafish proteins. Amino acid coloring indicates their predominant chemical property (blue: basic; red: acidic; green: polar; black: hydrophobic; purple: neutral). Logo generated with WebLogo [49].

**Figure 3 genes-11-00556-f003:**
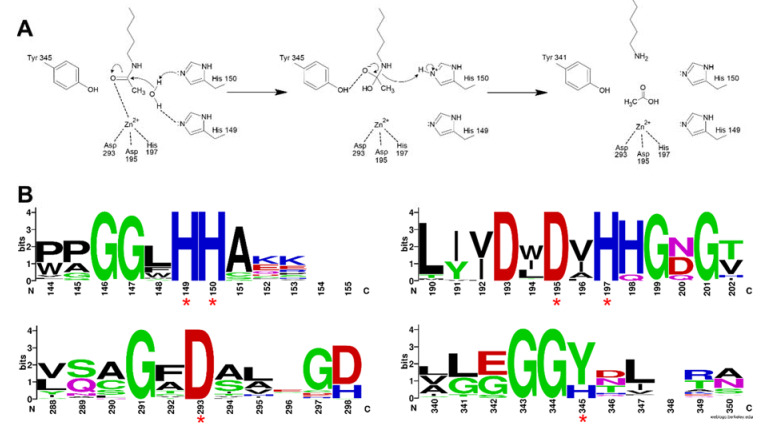
Catalytic action of HDACs. (**A**) Proposed mechanism of acetyl-lysine hydrolysis catalyzed by HDACs. (**B**) Sequence conservation of four regions containing the key catalytic amino acids (red asterisks) between all HDACs of the model species used for the phylogenetic tree in Figure 1. Residue Y345 is replaced by an H only in vertebrate Class IIa HDACs.

**Figure 4 genes-11-00556-f004:**
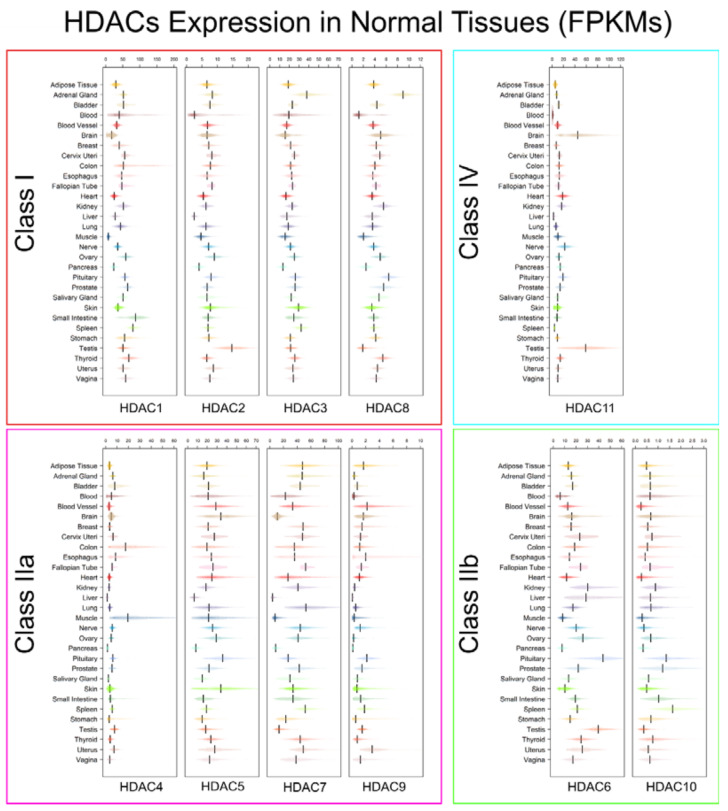
Transcript levels of HDACs 1–11 in healthy tissues expressed in FPKM (fragments per kilobase of transcript length per million of mapped reads). Boxplots show the expression of Class I (red box), Class IIa (pink box), Class IIb (green box), and Class IV (aquamarine box) from normal samples in the GTEx Release V8 dataset. Expression was FPKM-normalized using the length of ENSEMBL longest isoform and RNA-Seq data from GTEx [105].

**Figure 5 genes-11-00556-f005:**
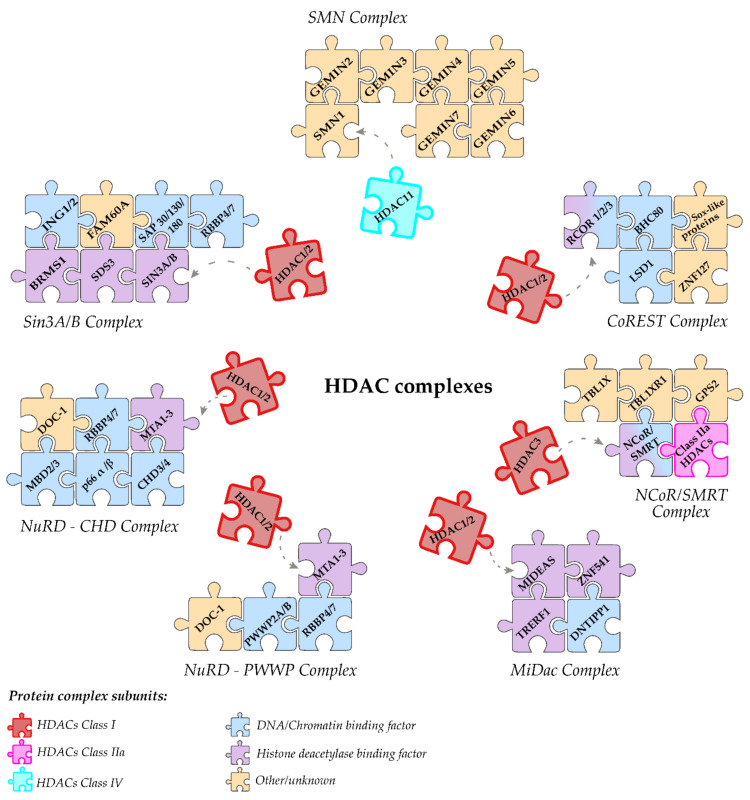
Schematic representation of the well characterized human HDAC-containing core complexes. Color schemes associated with HDACs subunits refer to sequence-based classification of HDACs described in this review. The recruitment of each HDAC in the core complex is depicted schematically with grey arrows that also identifying the HDAC-binding subunit. The HDAC1/2/11-binding subunit within each complex is indicated by the purple color. In addition, each complex contains multiple DNA/chromatin binding subunits indicated in blue. Finally, other subunits, where the biological function in each complex is still uncharacterized, are indicated by the sandy color. The puzzle piece representation and orientation does not reflect specific surface interaction regions. HDAC1/2 indicates 1 or 2, interchangeably (not as a heterodimer).

**Figure 6 genes-11-00556-f006:**
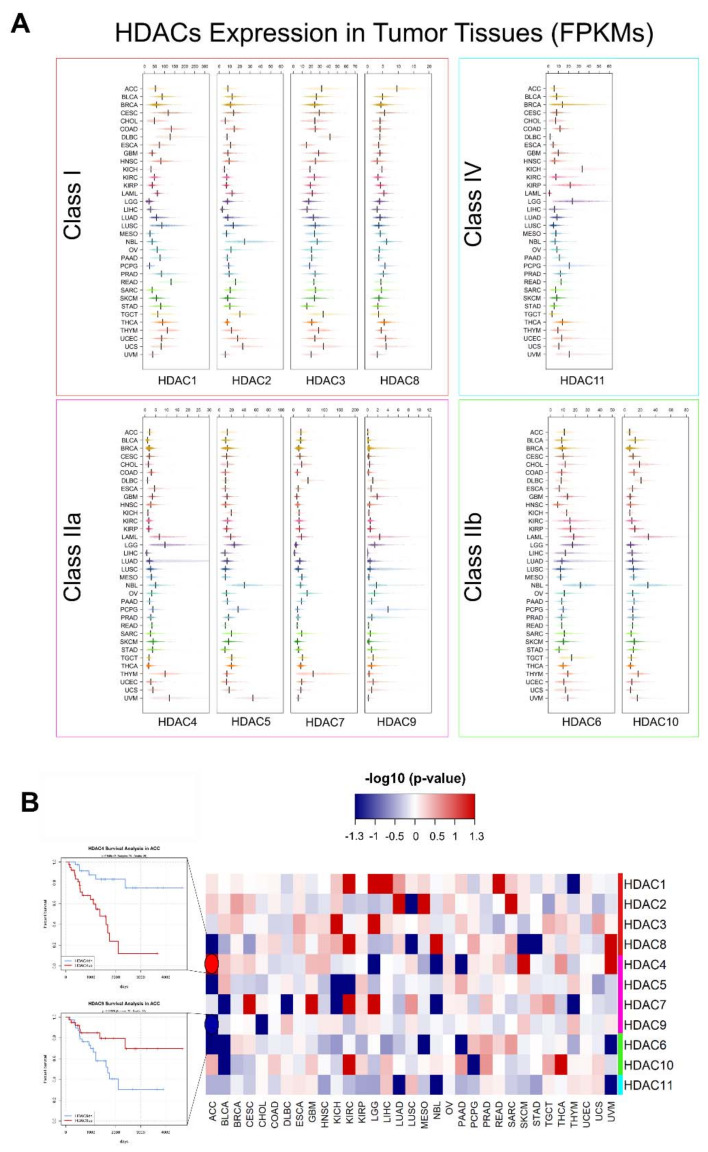
(**A**) Transcript levels of HDACs 1–11 in tumor tissues expressed in FPKM. Boxplots shows the expression of Class I (red box), Class IIa (pink box), Class IIb (green box), and Class IV (aquamarine box) from tumor samples in the pan-cancer TCGA dataset [187]. Tumor types are: ACC, adrenocortical carcinoma; BLCA, bladder urothelial carcinoma; BRCA, breast invasive carcinoma; CESC, cervical squamous cell carcinoma and endocervical adenocarcinoma; CHOL, cholangiocarcinoma; COAD, colon adenocarcinoma; DLBC, lymphoid neoplasm diffuse large B-cell lymphoma; ESCA, esophageal carcinoma; GBM, glioblastoma multiforme; HNSC, head and neck squamous cell carcinoma; KICH, kidney chromophobe; KIRC, kidney renal clear cell carcinoma; KIRP, kidney renal papillary cell carcinoma; LAML, acute myeloid leukemia; LGG, brain lower grade glioma; LIHC, liver hepatocellular carcinoma; LUAD, lung adenocarcinoma; LUSC, lung squamous cell carcinoma; MESO, mesothelioma; OV, ovarian serous cystadenocarcinoma; NBL, TARGET-neuroblastoma [188]; PAAD, pancreatic adenocarcinoma; PCPG, pheochromocytoma and paraganglioma; PRAD, prostate adenocarcinoma; READ, rectum adenocarcinoma; SARC, sarcoma; SKCM, skin cutaneous melanoma; STAD, stomach adenocarcinoma; TGCT, testicular germ cell tumors; THYM, thymoma; THCA, thyroid carcinoma; UCS, uterine carcinosarcoma; UCEC, uterine corpus endometrial carcinoma, UVM, uveal melanoma. Expression was FPKM-normalized using the length of ENSEMBL longest isoform and RNA-Seq data from TCGA. (**B**) Integrated HDAC survival analysis across tumors. Color intensity in the heatmap is proportional to -log10(p-value), a threshold of |1.3| corresponds to a *p*-value = 0.05. Red boxes in the heatmap correspond to a worse OS when the corresponding HDAC is upregulated, while blue boxes mean a better OS in case of upregulation, as shown by example survival curves (left side).

**Table 1 genes-11-00556-t001:** Classification of histone deacetylases. The current manuscript will focus on ‘classical’, Zn^2+^-dependent HDACs.

Family	Class (Yeast Homolog)	Subclass	Protein	Cell Compartment
Classical (Zn^2+^ dependent)	I (Rpd3)		HDAC1	Nucleus
			HDAC2	Nucleus
			HDAC3	Nucleus
			HDAC8	Nucleus
	II (Hda1)	IIa	HDAC4	Cytoplasm/Nucleus
			HDAC5	Cytoplasm/Nucleus
			HDAC7	Cytoplasm/Nucleus
			HDAC9	Cytoplasm/Nucleus
		IIb	HDAC6	Cytoplasm
			HDAC10	Cytoplasm
	IV (Rpd3, Hda1)		HDAC11	Cytoplasm/Nucleus
NAD dependent	III (Sir2, Hst1-4)		SIRT 1-7	Cytoplasm/Nucleus

**Table 2 genes-11-00556-t002:** Species selected for the phylogenetic analysis, with numbers of distinct HDAC genes detected. The analysis comprises a total of 223 protein sequences mapping to 25 organisms (24 *eukarya* and 1 *archaeon*). Three extra non-HDAC bacterial sequences from *B.subtilis*, *E.coli* and *A.aeolicus*, not indicated here but shown in Figure 1, were selected as outgroups to represent distance from the Bacteria kingdom.

Organism	Nr. of HDACs
*Homo sapiens* (human)	11
*Mus musculus* (mouse)	11
*Bos taurus* (cattle)	11
*Sus scrofa* (pig)	11
*Monodelphis domestica* (opossum)	10
*Ornithorhyncus anatinus* (platypus)	11
*Gallus gallus* (chicken)	10
*Tyto alba* (barn owl)	7
*Struthio camelus* (ostrich)	9
*Alligator sinensis* (alligator)	11
*Thamnophis elegans* (snake)	11
*Chelonia mydas* (turtle)	11
*Xenopus tropicalis* (frog)	11
*Danio rerio* (zebrafish)	11
*Latimeria chalumnae* (coealacanth)	11
*Callorhinchus milii* (shark)	9
*Anopheles gambiae* (mosquito)	5
*Drosophila melanogaster* (fruit fly)	5
*Centruroides sculpturatus* (scorpion)	7
*Caenorhabditis elegans* (nematode)	8
*Arabidopsis thaliana* (thale cress)	14
*Oryza sativa* (rice)	11
*Schizosaccharomices pombe* (fission yeast)	3
*Saccharomyces cerevisiae* (budding yeast)	3
*Pyrococcus furiosus* (archaeon)	1

**Table 3 genes-11-00556-t003:** List of HDAC inhibitors (HDACi). Columns indicate: structural class (as described in the text); name of the inhibitor (in brackets, alternative name); specificity of inhibition (‘Pan’ indicating an activity against all HDACs); cancer type in which the drug is currently being tested; references (with focus on drug tests in cancer contexts). The following drugs in this table have been approved by the Food & Drug Administration (FDA), as of May 2020: Vorinostat, Belinostat, Panobinostat, Valproic acid and Romidepsin.

Structural Group	Name	Specificity	Cancer	Reference
***Group I - Hydroxamic acids***				
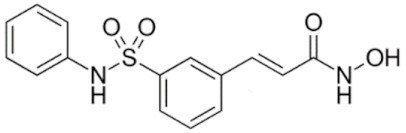	Trichostatin A (TSA)	Pan	Pancreatic, prostate, NSCLC, gastric	[265,266,267,268]
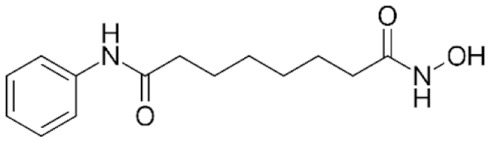	Vorinostat (SAHA)	Pan	Ovarian, gastric, leukemia	[221,268,269]
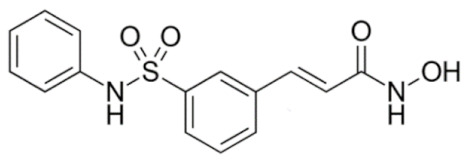	Belinostat (PXD101)	Pan	T-Cell lymphoma, pancreatic, hepatocellular carcinoma, acute myelogenous leukemia	[270,271,272,273,274]
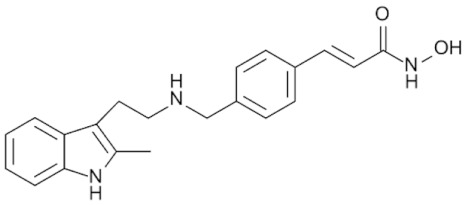	Panobinostat (LBH589)	Pan	MLL-ALL, neuroblastoma, esophageal squamous cell carcinoma, NSCLC, ovarian, pontine glioma, DLBCL	[216,223,227,275,276,277,278]
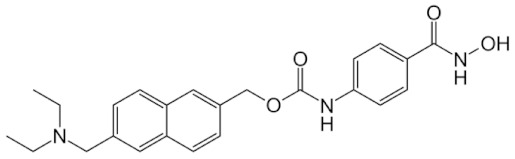	Givinostat (ITF-2357)	Pan	NSCLC, acute lymphoblastic leukemia, glioblastoma, acute lymphoblastic leukemia	[279,280,281,282]
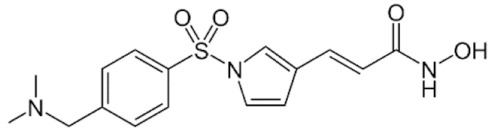	Resminostat (RAS2410)	Pan	Prostate, pancreas, NSCLC, hepatocellular carcinoma	[283,284,285,286]
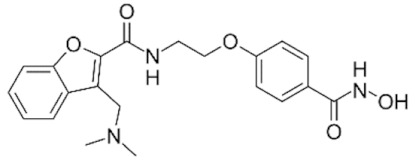	Abexinostat (PCI-24781)	Pan	breast, gastric, neuroblastoma, DLBCL	[267,287,288,289]
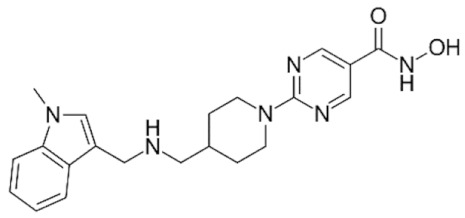	Quisinostat (JNJ-26481585)	Pan	Lung, multiple myeloma, rhabdomyosarcoma	[290,291,292]
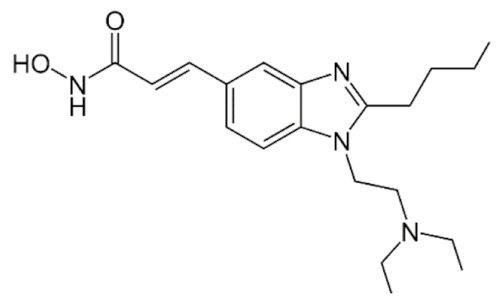	Pracinostat (SB939)	Pan	Acute lymphoblastic leukemia, myelodysplastic syndromes, myelofibrosis, NSCLC	[222,293,294,295]
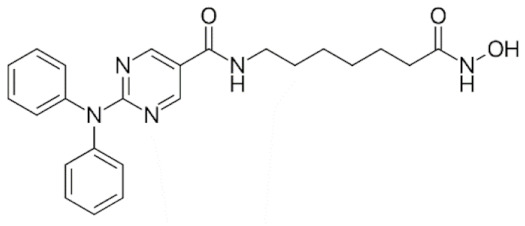	Rocilinostat (ACY-1215)	HDAC6	Esophageal squamous cell carcinoma, DLBCL, multiple myeloma, NSCLC	[229,296,297,298]
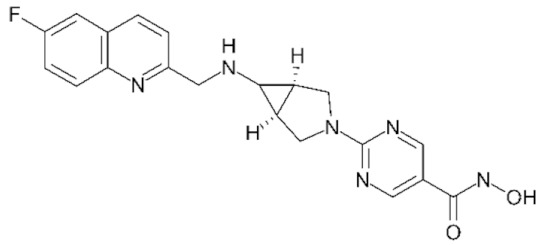	Nanatinostat (CHR-3996)	Class I & II	HCC, multiple myeloma	[228,299,300]
***Group II—SCFAs***				
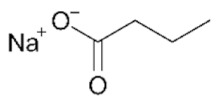	Sodium butyrate and derivatives	Class I & IIa	Breast, colorectal, gastric	[235,268,301]
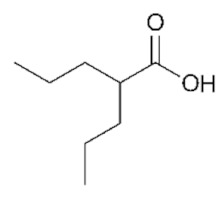	Valproic acid (VPA)	Class I	CNS, colorectal, prostate cancer, NSCLC, pancreatobiliary tract	[302,303,304,305]
***Group III—Benzamides***				
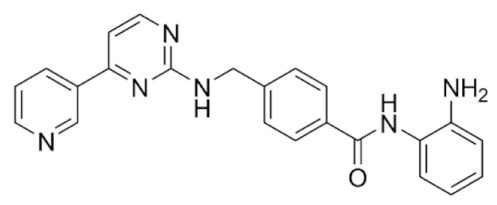	Mocetinostat (MGCD0103)	Class I & IV	Pancreatic, Hodgkin’s lymphoma, SHH medulloblastoma, leiomyosarcoma, prostate, glioblastoma, ovarian	[197,306,307,308,309,310,311]
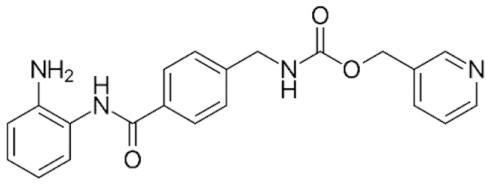	Entinostat (MS-275)	Class I & IV	Breast, leukemia, ovarian, renal cell carcinoma	[244,245,312,313]
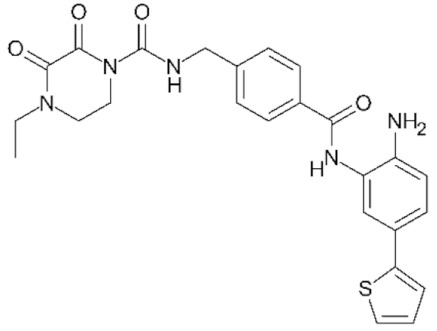	K560 and derivatives	Class I	N/A	[248,249]
***Group IV—Tetrapeptides*** **(depsipeptides)**				
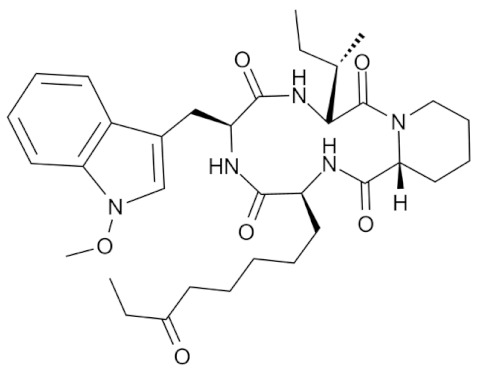	Apicidin (OSI-2040)	Class I	Oral squamous cell carcinoma, ovarian, pancreatic	[264,314,315]
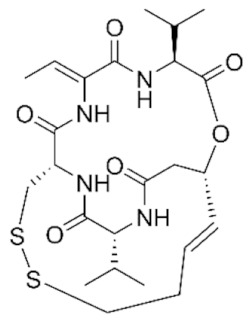	Romidepsin (FK228)	Class I	Germ cell, HCC, T cell lymphoma	[256,257,259]
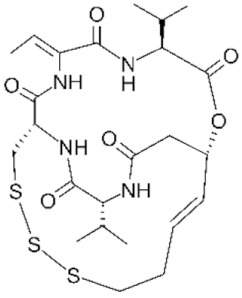	Chromopeptide A	Class I	Prostate	[255]
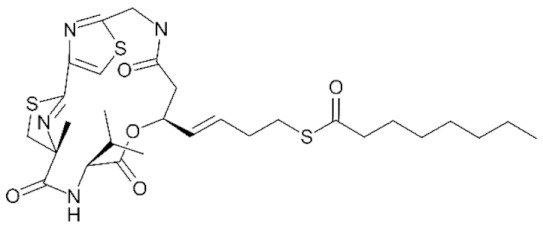	Largazole	Class I	Colon, lung	[316,317,318]
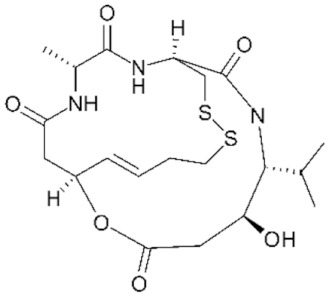	Spiruchostatin A	Class I	Bladder, breast	[319,320]
***Tetrapeptides*** **(** **α** **4-cyclotetrapeptides)**				
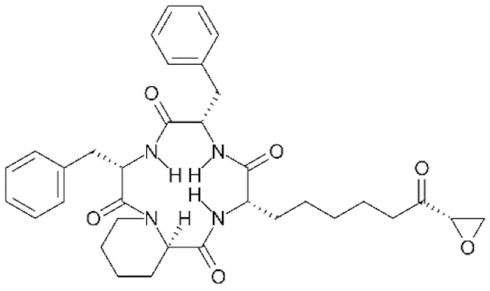	Trapoxin	HDAC8	N/A	[252,261]
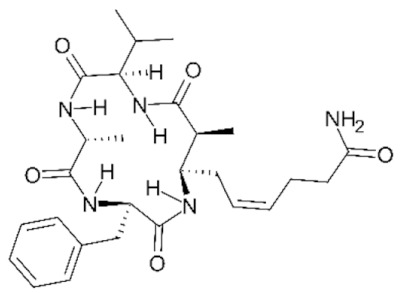	Azumamide A and derivates	Class I	N/A	[321]
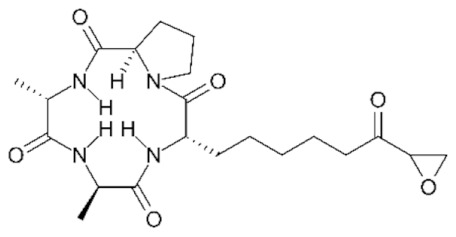	HC-toxin	Class I	Cholangiocarcinoma, breast, neuroblastoma, medulloblastoma	[322,323,324,325]

Molecule structures in Table 3 were obtained from PubChem [326] or from the indicated references, and drawn using MarvinSketch [327].

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
