# Peer review of "Histone Deacetylases (HDACs): Evolution, Specificity, Role in Transcriptional Complexes, and Pharmacological Actionability"

_genes, 2020, doi:10.3390/genes11050556_

Round 1
Reviewer 1 Report
The review entitled ‘Histone Deacetylases (HDACs): evolution, specificity, role in transcriptional complexes and pharmacological actionability’ by Giorgio Milazzo at al., is overall very well written. This is certainly a timely topic given the use of HDAC as anti-cancer drugs and will therefore be interesting for researchers to read.
The discoveries of HDACs are nicely introduced from a historical perspective. This is followed by a phylogenetic analysis of 226 representatives of the HDAC protein family and a representation of the different domains of the mammalian HDACs within each family. After analysing the molecular mechanism of histone deacetylation, Milazzo et al deeply analyse the expression patterns among different normal and tumoral tissues and focus on the different HDAC complexes described and a vast analysis of their functions, particularly regarding gene expression.
However, I feel that the review is too focused on the consequences of HDAC on transcription (as correctly pointed out already in the title) and I have missed the mention of the known role of acetylation in other DNA metabolic processes such as DNA replication or DNA damage repair which are major contributors to genome stability. This is of particular relevance given the focus of the review on cancer, a genetic instability associated disease, among human pathogenesis.
I realised that the role of HDACs on non-histone targets has been covered even when not related to transcription (for instance, on Page 22 line 783 onwards, the authors explain the role of HDAC11 in Cdt1 de-acetylation). However, the readers might get the wrong impression that the de-acetylation of histones is only involved in regulating gene expression. Although transcription regulation is a major role of histone de-acetylation and it is for which there is more literature available, the authors should at least mention the fact that the de-acetylation of histones, as well as many other PTMs, is of relevance for other metabolic processes and that this might account for some of the pathogenic and therapeutic roles of HDAC inhibitors. If the authors do not want to further extend this already extensive review, this could be mentioned in the beginning of the paper and state that, however, this review will rather focus transcription.
Along this line, I believe that naming HDAC complexes instead of HDAC transcriptional complexes in figure 5 could be more accurate.
Other comments:
page 2 line 69 onwards: ‘Yeast studies showed that Hda1 plays a more prominent role in regulating carbon metabolite and carbohydrate transport and utilization, while Rpd3 is involved in cell cycle progression’ and page 3 line 91: ‘the yeast Sir2 protein (Table 1), an HDAC homolog involved in
92 amino acid biosynthesis’. These sentences give the wrong impression that HDACs are directly involved in these processes whereas I would say that, for, instance Sir2 is mainly involved in gene silencing in yeast. I think that clarifying that these are the main genes regulated by these enzymes as shown by bioinformatics analyses is necessary. A similar issue happens on page 22, line 768: ‘The predominant biological processes attributable to HDAC11 are relative to the immune system responses, oligodendrocyte development, and mRNA splicing.’
Page 7, line 191: ‘Class IIa HDACs are characterized by a catalytic domain occupying the C‐terminal half of the protein, and by a N‐terminal part rich in interaction domains, including with the MEF2 transcription factor, a master regulator of muscle organogenesis’. If HDAC5 does not have the MEF2 interaction domain as it appears in Figure 2A, I suggest to re-structure this sentence.
Figure 2B refers to the aa enrichments of the Class II HDACs. What is the logo for the NLS in the other families (class I and class IV)?
Figure 3B is not mentioned in the text. It is not clear what it is about. What is the difference between the four depicted regions?
Line 376 and legend of figure 4, please state that FPKM states from Fragments Per Kilobase Million.
Page 22, line 783, HDAC11 instead of HDAC 11.
Some lines in Table 3 seem to have been lost (see for instance the lines between the Belinostat and Panobinostat chemical formulae).
Author Response
We have received the reviewers’ comments to our review describing the role, specificity and evolution of HDACs (manuscript id genes-805319), destined to the Special Issue “Evolution of Gene Regulatory Networks”. We have addressed all comments, as reported below. This has also allowed us to polish the manuscript, fix typos, improve the quality of the tables and extend considerations on the roles of HDACs beyond transcriptional regulation. We hope the manuscript will meet the criteria for publication on Genes.
Reviewer 1
The review entitled ‘Histone Deacetylases (HDACs): evolution, specificity, role in transcriptional complexes and pharmacological actionability’ by Giorgio Milazzo at al., is overall very well written. This is certainly a timely topic given the use of HDAC as anti-cancer drugs and will therefore be interesting for researchers to read.
The discoveries of HDACs are nicely introduced from a historical perspective. This is followed by a phylogenetic analysis of 226 representatives of the HDAC protein family and a representation of the different domains of the mammalian HDACs within each family. After analysing the molecular mechanism of histone deacetylation, Milazzo et al deeply analyse the expression patterns among different normal and tumoral tissues and focus on the different HDAC complexes described and a vast analysis of their functions, particularly regarding gene expression.
Answer: We thank the reviewer for these encouraging observations. We did our best in merging the diverse expertise of our lab, both computational and experimental.
However, I feel that the review is too focused on the consequences of HDAC on transcription (as correctly pointed out already in the title) and I have missed the mention of the known role of acetylation in other DNA metabolic processes such as DNA replication or DNA damage repair which are major contributors to genome stability. This is of particular relevance given the focus of the review on cancer, a genetic instability associated disease, among human pathogenesis. I realised that the role of HDACs on non-histone targets has been covered even when not related to transcription (for instance, on Page 22 line 783 onwards, the authors explain the role of HDAC11 in Cdt1 de-acetylation). However, the readers might get the wrong impression that the de-acetylation of histones is only involved in regulating gene expression. Although transcription regulation is a major role of histone de-acetylation and it is for which there is more literature available, the authors should at least mention the fact that the de-acetylation of histones, as well as many other PTMs, is of relevance for other metabolic processes and that this might account for some of the pathogenic and therapeutic roles of HDAC inhibitors. If the authors do not want to further extend this already extensive review, this could be mentioned in the beginning of the paper and state that, however, this review will rather focus transcription. Along this line, I believe that naming HDAC complexes instead of HDAC transcriptional complexes in figure 5 could be more accurate.
Answer: We agree with the reviewer that our review is focused on transcription, as the title indicates. This was a precise strategic choice on our side, since focusing also on non-transcriptional roles of HDACs would have increased even further the size and scope of the manuscript, which already has more than 300 citations. However, we concur that in the previous form, the manuscript omitted to clearly mention that HDACs have established roles in other DNA metabolic processes, which are also important factors in disease (e.g. through genomic instability). We therefore expanded our text, (specifically in paragraph 3.1, page 8 and in the Conclusions, page 36), to discuss roles of HDACs beyond regulation of transcription. This also allowed us to highlight a previously unknown role of HDACs in the pathogenesis of SARS-CoV-2 infection, recently highlighted by the Gordon et al., 2020 paper on Nature (Pubmed id 32353859), on the interactions between viral and human proteins. We have now mentioned early in the text that the focus of the review is on transcription. Finally, we have modified Figure 5 as requested.
Other comments:
page 2 line 69 onwards: ‘Yeast studies showed that Hda1 plays a more prominent role in regulating carbon metabolite and carbohydrate transport and utilization, while Rpd3 is involved in cell cycle progression’ and page 3 line 91: ‘the yeast Sir2 protein (Table 1), an HDAC homolog involved in 92 amino acid biosynthesis’. These sentences give the wrong impression that HDACs are directly involved in these processes whereas I would say that, for, instance Sir2 is mainly involved in gene silencing in yeast. I think that clarifying that these are the main genes regulated by these enzymes as shown by bioinformatics analyses is necessary. A similar issue happens on page 22, line 768: ‘The predominant biological processes attributable to HDAC11 are relative to the immune system responses, oligodendrocyte development, and mRNA splicing.’
Answer: We clarified the text, describing the direct role of Hda1, Rpd3, Sir2 and HDAC11 in transcriptional regulation (and therefore only indirectly in the aforementioned pathways).
Page 7, line 191: ‘Class IIa HDACs are characterized by a catalytic domain occupying the C‐terminal half of the protein, and by a N‐terminal part rich in interaction domains, including with the MEF2 transcription factor, a master regulator of muscle organogenesis’. If HDAC5 does not have the MEF2 interaction domain as it appears in Figure 2A, I suggest to re-structure this sentence.
Answer: We thank the reviewer for noticing this omission. We added the MEF2 interaction domain in the cartoon for the HDAC5 protein.
Figure 2B refers to the aa enrichments of the Class II HDACs. What is the logo for the NLS in the other families (class I and class IV)?
Answer: The NLS of Class II HDAC is the most studied and, we believe, the only one with enough experimental support to warrant a conservation analysis. This is the reason why we omitted an in-depth analysis of NLS from other Classes.
Figure 3B is not mentioned in the text. It is not clear what it is about. What is the difference between the four depicted regions?
Answer: Our most sincere thanks for noticing this. Figure 3 B was mistakenly neither cited nor described in the text. We added a few sentences to correct this mistake, as we believe it’s important to highlight the Y345 amino acid as being the only core catalytical element not fully conserved in all HDACs.
Line 376 and legend of figure 4, please state that FPKM states from Fragments Per Kilobase Million.
Answer: We expanded the acronym as requested.
Page 22, line 783, HDAC11 instead of HDAC 11.
Answer: We corrected this typo.
Some lines in Table 3 seem to have been lost (see for instance the lines between the Belinostat and Panobinostat chemical formulae).
Answer: Thank you for noticing this. We fear that the line loss was a problem of the PDF rendering of the manuscript. We rehauled Table 3 as per request of reviewer 2 and double-checked the hydroxamic acid group in the current version.
Reviewer 2 Report
Giorgio Milazzo et al. propose a very comprehensive and exhaustive review on HDACs (in Homo sapiens and other organisms, their specificity, their role in Transcriptional complexes as well as the Pan‐cancer analysis of them). Then, the authors describe HDAC inhibitors and their potentials as innovative therapeutics as anti-cancer agents.
The authors have a great expertise on these topics and this review will undoubtedly contribute to the field. I would recommend this review for publication in Genes.
I have only some minor comments:
1) The HDACI classification is a bit more complicated as those described by the authors. The authors miss the description of a new classification proposed by Melesina et al. (Journal of Molecular Graphics and Modelling 62 (2015) 342–361, FutureMed. Chem 2018).
2) The authors say at the beginning of their review « We will focus on the largest family of histone deacetylases, Zinc dependent HDACs… » (p1 l33), so why do they described the « (V) sirtuin inhibitors such as: Nicotinamide (Pan‐inhibitor) and the SIRT1 and SIRT2 specific inhibitors Sirtinol and Cambinol, respectively [180,185,200].
Moreover, the inhibitors are not described in the Table 3. I suggest to remove the sentence. Then, to modify :
p26 l945: HDACi are also classified into fourth instead of HDACi are also classified into fifth
3) Concerning the Table 3:
- The drawings of the molecules are awful. I have never seen atoms at an angle. The size of the atoms is not homogeneous, even if the molecules are large, an effort can be made to make them more readable.
- It is not necessary to keep the column « FDA Approved » for only four FDA approved compounds. Please, remove the column, it is already described in the text. Then, there will be more place for the compounds.
- P27 l950: Please, remove « the clinical status » in the sentence The family/class/HDAC specificity, the clinical status and the types of cancer/disease
- p35 l1148: Please, remove the sentence « approval by Food & Drug Administration for pharmacological use in humans »
- Some compounds have no description on their target/implication in cancer, some cells are empty in the table 3, I was wondering if it would be more judicious to remove them or to complete the table. I mean if there is no relation between the described compound and an activity on cancer field, it is not necessary to keep them, because the interest of the Table is precisely this relation.
4) There are a number of typos and grammar errors. A few examples are included here (but not limited to them):
P1 l11 in Abstract: acetyl instead of acetyle
p7 l203: Arginine instead of argynines (R)
p8 l240: zinc ion (Zn2+) instead of zinc ion (Zn2+) , 2+ in superscript
p8 l242: The Zn2+ cation instead of The Zn2+ cation, 2+ in superscript
p8 l243: making the carbonyl carbon more electrophilic instead of making the carbonyl carbon a better target
p9 l265: Tyrosine instead of tyrosine
p10 l312 and l324: Class IIa HDACs instead of class IIa HDACs
It will be interesting to be homogeneous for upper or lower case everywhere for “class” or “aa name”.p10 l320: to deacetylate instead of to deacetylatesp11 l347: responsible for promoting instead of responsible to promotingp11 l353: Class I HDACs instead of class I HDACs
IFN has to be in the « Abbreviations » list
p13 l382: HDAC2 expression varies instead of HDAC2 expression vary
p16 l384: HDACs subunits refer to a sequence‐based instead of HDACs subunits refers to sequence‐based
p16 l385: The recruitment of each HDAC instead of The recruitment of the each HDAC
p17 l537: Mesenchimal Transition instead of Mesenchimal Trasition
p19 l628: PWWP2A and PWWP2B instead of PWWP2A and PWWP2
p19 l635: BEND325 allows for the instead of BEND325 allows for the the
p22 l772: it is composed of six protein instead of it is composed by six protein
p22 l786: Lys49 at instead of Lys49at one, space is missing
p24 l818: blue boxes instead of blu boxes
p26 l946: main groups depending on instead of main groups depend on
p27 l957: Zn2+ cation instead of Zn2+ cation, 2+ in superscript and p30 l1114 idem
p27 l968: non-Hodgkin instead of non‐Hodgikn
p27 l992: IC50, 50 has to be in underscript
p28 l1030: 4‐carbon atoms chain instead of 4‐carbon
p29 l1052: phase I‐III for several instead of phase I‐III for for several
p30 l1109: bicyclic instead of bycyclic
p30 l1140: via should be in italic
Author Response
Reviewer 2
Giorgio Milazzo et al. propose a very comprehensive and exhaustive review on HDACs (in Homo sapiens and other organisms, their specificity, their role in Transcriptional complexes as well as the Pan‐cancer analysis of them). Then, the authors describe HDAC inhibitors and their potentials as innovative therapeutics as anti-cancer agents. The authors have a great expertise on these topics and this review will undoubtedly contribute to the field. I would recommend this review for publication in Genes.
Answer: We are grateful for these encouraging statements and we sincerely hope this review will be useful to the field.
I have only some minor comments:
1) The HDACI classification is a bit more complicated as those described by the authors. The authors miss the description of a new classification proposed by Melesina et al. (Journal of Molecular Graphics and Modelling 62 (2015) 342–361, FutureMed. Chem 2018).
Answer: We thank the reviewer for pointing out this paper that we missed in the first version of the manuscript. We used the currently accepted classification that is predominant in the field of oncology, however we added a few sentences to specify that more classification systems exist, and that HDAC inhibitors can have further targets beyond cancer, such parasitic HDACs (Melesina et al., mentioned by the reviewer).
2) The authors say at the beginning of their review « We will focus on the largest family of histone deacetylases, Zinc dependent HDACs… » (p1 l33), so why do they described the « (V) sirtuin inhibitors such as: Nicotinamide (Pan‐inhibitor) and the SIRT1 and SIRT2 specific inhibitors Sirtinol and Cambinol, respectively [180,185,200]. Moreover, the inhibitors are not described in the Table 3. I suggest to remove the sentence. Then, to modify: p26 l945: HDACi are also classified into fourth instead of HDACi are also classified into fifth
Answer: We apologize for this oversight. As the review is not focused on sirtuins, we have removed the sentence.
3) Concerning the Table 3:
The drawings of the molecules are awful. I have never seen atoms at an angle. The size of the atoms is not homogeneous, even if the molecules are large, an effort can be made to make them more readable.
Answer: We have to concur with the reviewer, the molecules were poorly shown. We redrew them based on PubChem structures and using a dedicated software (MarvinSketch).
It is not necessary to keep the column « FDA Approved » for only four FDA approved compounds. Please, remove the column, it is already described in the text. Then, there will be more place for the compounds.
Answer: Excellent suggestion. We removed the column and transferred the information on FDA approval to the text.
P27 l950: Please, remove « the clinical status » in the sentence The family/class/HDAC specificity, the clinical status and the types of cancer/disease
Answer: Thanks for the suggestion, this modification made the sentence more correct and more readable.
p35 l1148: Please, remove the sentence « approval by Food & Drug Administration for pharmacological use in humans »
Answer: The sentence has now been removed from the legend of Figure 3.
Some compounds have no description on their target/implication in cancer, some cells are empty in the table 3, I was wondering if it would be more judicious to remove them or to complete the table. I mean if there is no relation between the described compound and an activity on cancer field, it is not necessary to keep them, because the interest of the Table is precisely this relation.
Answer: We thank the reviewer for noticing this. We believe Table 3 had some formatting errors in the previous version of the manuscript. We have corrected this now.
4) There are a number of typos and grammar errors. A few examples are included here (but not limited to them):
P1 l11 in Abstract: acetyl instead of acetyle
p7 l203: Arginine instead of argynines (R)
p8 l240: zinc ion (Zn2+) instead of zinc ion (Zn2+) , 2+ in superscript
p8 l242: The Zn2+ cation instead of The Zn2+ cation, 2+ in superscript
p8 l243: making the carbonyl carbon more electrophilic instead of making the carbonyl carbon a better target
p9 l265: Tyrosine instead of tyrosine
p10 l312 and l324: Class IIa HDACs instead of class IIa HDACs
It will be interesting to be homogeneous for upper or lower case everywhere for “class” or “aa name”.
p10 l320: to deacetylate instead of to deacetylates
p11 l347: responsible for promoting instead of responsible to promoting
p11 l353: Class I HDACs instead of class I HDACs
IFN has to be in the « Abbreviations » list
p13 l382: HDAC2 expression varies instead of HDAC2 expression vary
p16 l384: HDACs subunits refer to a sequence‐based instead of HDACs subunits refers to sequence‐based
p16 l385: The recruitment of each HDAC instead of The recruitment of the each HDAC
p17 l537: Mesenchimal Transition instead of Mesenchimal Trasition
p19 l628: PWWP2A and PWWP2B instead of PWWP2A and PWWP2
p19 l635: BEND325 allows for the instead of BEND325 allows for the the
p22 l772: it is composed of six protein instead of it is composed by six protein
p22 l786: Lys49 at instead of Lys49at one, space is missing
p24 l818: blue boxes instead of blu boxes
p26 l946: main groups depending on instead of main groups depend on
p27 l957: Zn2+ cation instead of Zn2+ cation, 2+ in superscript and p30 l1114 idem
p27 l968: non-Hodgkin instead of non‐Hodgikn
p27 l992: IC50, 50 has to be in underscript
p28 l1030: 4‐carbon atoms chain instead of 4‐carbon
p29 l1052: phase I‐III for several instead of phase I‐III for for several
p30 l1109: bicyclic instead of bycyclic
p30 l1140: via should be in italic
Answer: We thank the reviewer for his/her deep scrutiny and attention to detail. This allowed us to fix all the mentioned typos, as well as other minor typos identified in an in-depth proofreading.